# X chromosome dosage drives statin-induced dysglycemia and mitochondrial dysfunction

Peixiang Zhang[1,10], Joseph J. Munier[2], Carrie B. Wiese[1], Laurent Vergnes[1], Jenny C. Link [3,11], Fahim Abbasi [4], Emilio Ronquillo[1], Katherine Scheker[1], Antonio Muñoz [5], Yu-Lin Kuang [5], Elizabeth Theusch [5], Meng Lu[6], Gabriela Sanchez[6], Akinyemi Oni-Orisan [7], Carlos Iribarren[6], Michael J. McPhaul[8], Daniel K. Nomura [9], Joshua W. Knowles [4], Ronald M. Krauss[5], Marisa W. Medina [5] & Karen Reue [1,3] ✉

Statin drugs lower blood cholesterol levels for cardiovascular disease prevention. Women are more likely than men to experience adverse statin effects, particularly new-onset diabetes (NOD) and muscle weakness. Here we find that impaired glucose homeostasis and muscle weakness in statin-treated female mice are associated with reduced levels of the omega-3 fatty acid, docosahexaenoic acid (DHA), impaired redox tone, and reduced mitochondrial respiration. Statin adverse effects are prevented in females by administering fish oil as a source of DHA, by reducing dosage of the X chromosome or the *Kdm5c* gene, which escapes X chromosome inactivation and is normally expressed at higher levels in females than males. As seen in female mice, we find that women experience more severe reductions than men in DHA levels after statin administration, and that DHA levels are inversely correlated with glucose levels. Furthermore, induced pluripotent stem cells from women who developed NOD exhibit impaired mitochondrial function when treated with statin, whereas cells from men do not. These studies identify X chromosome dosage as a genetic risk factor for statin adverse effects and suggest DHA supplementation as a preventive co-therapy.

Statins, or HMG-CoA reductase enzyme inhibitors, are the most widely prescribed drug class for reducing blood cholesterol levels and risk of cardiovascular disease[1]. However, statins may have adverse effects, the most common of which are myopathy, which may occur in 5–10% of statin users[2], and new-onset diabetes[3–7]. There are significant data showing that statin adverse effects are more prevalent in women than in men. A recent meta-analysis of 176 statin trials with >4 million subjects identified female sex as the strongest risk factor for statin intolerance, in which myopathy was a major outcome (OR 1.47, $p = 0.007$)[8]. Female sex had a greater impact on statin adverse effects than ethnic background, age, obesity, diabetes, or other chronic conditions. Myopathy was also shown to be 2-fold more prevalent in women

[1]Human Genetics, David Geffen School of Medicine, University of California, Los Angeles, CA 90095, USA. [2]Molecular, Cellular & Integrative Physiology, University of California, Los Angeles, CA, USA. [3]Molecular Biology Institute, University of California, Los Angeles, CA, USA. [4]Division of Cardiovascular Medicine and Cardiovascular Institute, Diabetes Research Center, Stanford University School of Medicine, Stanford, CA, USA. [5]Department of Pediatrics, University of California, San Francisco, Oakland, CA, USA. [6]Division of Research, Kaiser Permanente, Oakland, CA, USA. [7]Institute for Human Genetics, University of California, San Francisco, CA, USA. [8]Quest Diagnostics Nichols Institute, San Juan Capistrano, CA 92675, USA. [9]Nutritional Sciences and Toxicology, and Novartis-Berkeley Center of Proteomics and Chemistry Technologies, University of California, Berkeley, Berkeley, CA, USA. [10]Present address: School of Medicine, University of Maryland, Baltimore, MD, USA. [11]Present address: Department of Biology, Whittier College, Whittier, CA, USA. ✉e-mail: reuek@ucla.edu

compared to men in a separate meta-analysis[9], and assessment of incident diabetes in statin clinical trials with stratification by sex have shown greater diabetes risk in women compared to men[10,11].

We exploited the sex differences in susceptibility to statin-associated myopathy and new-onset diabetes (NOD) to identify mechanisms that contribute to these adverse effects. We used mouse models that allow comparison of males and females on the same genetic background, as well as analysis of the contribution of gonadal sex differences (ovaries vs. testes) and chromosomal sex differences (XX vs. XY), to identify molecular and genetic determinants of statin-related impairment of glucose homeostasis and muscle strength.

We determined that statin drug interacts with female sex, particularly XX sex chromosome complement, to alter the lipidome and transcriptome in ways that impact mitochondrial function and cellular redox tone. We further identified nutritional and genetic manipulations that prevent statin adverse effects in female mice. Ultimately, we verified the findings from mice in human cohorts, showing differential alterations in lipid metabolism and mitochondrial function in women compared to men treated with statin. Our findings suggest a potential nutritional co-therapy to statins that may prevent adverse effects in vulnerable individuals.

## Results

### Female sex promotes statin-induced glucose intolerance and reduced muscle strength in the mouse

We treated cohorts of male and female C57BL/6 J mice (aged 10 weeks) with a chow diet without or with simvastatin (equivalent to human dose of 80 mg/day; see Methods). Inclusion of statin in the chow diet did not alter food intake, body weight, or body weight gain in male or female mice (Fig. 1a−c). Fasting glucose and glucose tolerance were assessed in independent cohorts at 2, 4, 8, and 16 weeks of statin treatment. Males had higher basal glucose levels, but statin treatment led to elevations in fasting glucose levels exclusively in females (Fig. 1d). Females developed significant glucose intolerance beginning at 4 weeks of statin treatment, which persisted throughout 16 weeks (Fig. 1e, Suppl. Fig. 1a). Statin-treated male mice also experienced impaired glucose tolerance, but this was delayed relative to female mice, occurring only after 8 or 16 weeks of statin treatment (Fig. 1d, Suppl. Fig. 1b). Statin treatment did not alter insulin levels or HOMA-IR in male or female mice, but male mice had higher absolute insulin levels and HOMA-IR compared to females (Suppl. Fig. 1c, d).

To assess whether mice experienced muscle pain or weakness that are characteristic of statin-related myopathy, we measured grip strength in the forelimbs with a grip bar strength dynamometer. Male mice had similar grip strength force on chow and statin, but female mice experienced a 20% reduction in grip strength after 8 weeks of statin treatment (Fig. 1f). We also noted sex differences in mitochondrial content in skeletal muscle due to sex and statin treatment. Mitochondrial copy number was assessed as the relative levels of mitochondrial DNA to genomic DNA determined by qPCR[12]. Under basal chow diet, female mice had lower muscle mitochondrial DNA copy number than males, and females significantly decreased mitochondrial number further in response to statin treatment (Fig. 1g).

### Statin adverse effects in female mice are associated with reduced ω-3 fatty acid levels

To identify potential sex differences in statin metabolism, we performed metabolic profiling of liver (the primary site of statin metabolism), skeletal muscle, and plasma by mass spectrometry. Minimal effects of statin treatment on metabolite concentrations were observed in either sex in muscle or plasma (Supplementary Data 1). However, statin altered hepatic lipids and other metabolites, with key sex differences (Fig. 2a). As expected, statin-treated males and females both experienced reduced levels of hepatic cholesterol species (females had reduced free cholesterol, and males reduced cholesterol

esters), and both showed trends to very modest reductions in mevalonate pathway intermediates (females reduced geranylgeranyl pyrophosphate, and males trended to reduced $CoQ_{10}$ and farnesyl pyrophosphate) (Suppl. Fig. 2a).

We hypothesized that statin-related adverse effects are most likely associated with changes that occur within sex in response to statin. Statin treatment caused a reduction in hepatic levels of several fatty acid species exclusively in females. These included a 30% reduction in medium- and long-chain saturated and unsaturated fatty acid species (C16:0, C18:1, C20:4) and a 40% reduction in the essential ω-3 polyunsaturated fatty acid, docosahexaenoic acid (DHA, C22:6) (Fig. 2c and Supplementary Data 1). The ω-3 fatty acid eicosapentaenoic acid (EPA, C20:5) was not detectable in our liver samples. Sex-specific alterations in the hepatic transcriptome were also observed in response to statin (Fig. 2b). Female-specific reductions occurred in the expression of genes required for fatty acid synthesis (Acly, Acaca, Acacb, Fasn), for desaturation of medium-chain fatty acids (Scd1), and for synthesis of ω-3 fatty acids (Elovl6) compared to their basal expression levels (Fig. 2d, Suppl. Fig. 2b).

Of the fatty acids reduced in statin-treated females, ω-3 fatty acid levels have previously been associated with glucose homeostasis, muscle health, reduced inflammation, and mitochondrial function[13–20]. We hypothesized that the 40% reduction in DHA levels in statin-treated females may contribute to the observed sex-biased adverse effects. In support of a role for DHA in preventing adverse statin effects, cultured mouse hepatocytes exposed to statin (10 μM simvastatin for 24 hr) had reduced viability that could be prevented by the addition of DHA, but not addition of saturated or monounsaturated fatty acids (Fig. 2e).

### Statin adverse effects in females is prevented by co-therapy with a source of ω-3 fatty acids

Our data raised the possibility that preventing a reduction in DHA levels in female mice treated with statin might diminish adverse effects. To test this hypothesis, we fed male and female mice the control or statin-containing chow and provided either fish oil (a source of ω-3 fatty acids, DHA and EPA) in coconut oil vehicle, or coconut oil vehicle alone. Lipidomic analysis of our fish and coconut oils confirmed that the fatty acid species present at higher levels in fish oil are ω-3 fatty acids (DHA, EPA, and docosapentaenoic acid, C22:5) (Suppl. Fig. 3). Fish/coconut oil dosing was performed by oral gavage 5 times/week. After 5 weeks, glucose tolerance and grip strength were determined. Fish oil co-therapy with statin protected female mice from impaired glucose tolerance (Fig. 3a) and prevented statin-induced reduction in grip strength (Fig. 3b, compare shaded yellow to shaded red bar). Fish oil had no effect on these traits in males (Fig. 3a, b, right panels).

To assess potential effects of statin and fish oil on glucose metabolism, we assessed glycogen content in liver and skeletal muscle. Female and male mice have similar basal levels of glycogen in liver and muscle, but only females experienced a ~50% reduction in glycogen levels in both tissues in response to statin treatment plus coconut oil vehicle (Fig. 3c, compare shaded red to solid red bars). Administration of fish oil in coconut oil vehicle prevented the decrement in hepatic and muscle glycogen levels in statin-treated females (Fig. 3c, compare shaded yellow to shaded red bars). No effects of statin or oil treatments were observed in males (Fig. 3c, right panels). The reduced glycogen levels in statin-treated females were associated with reduced phosphorylation of glycogen synthase kinase 3β, which corresponds to reduced glycogen synthesis and storage. We did not find evidence for increased hepatic gluconeogenesis or increased glycogenolysis gene expression in response to statins in either sex; in fact, females exhibited reduced expression of the gene for glycogen phosphorylase (Pygl), a key step in glycogenolysis, and males showed reduced expression of one isoform of phosphoglucomutase (Pgm3) (Suppl. Fig. 4). Our results indicate that sex-biased effects of glucose storage as

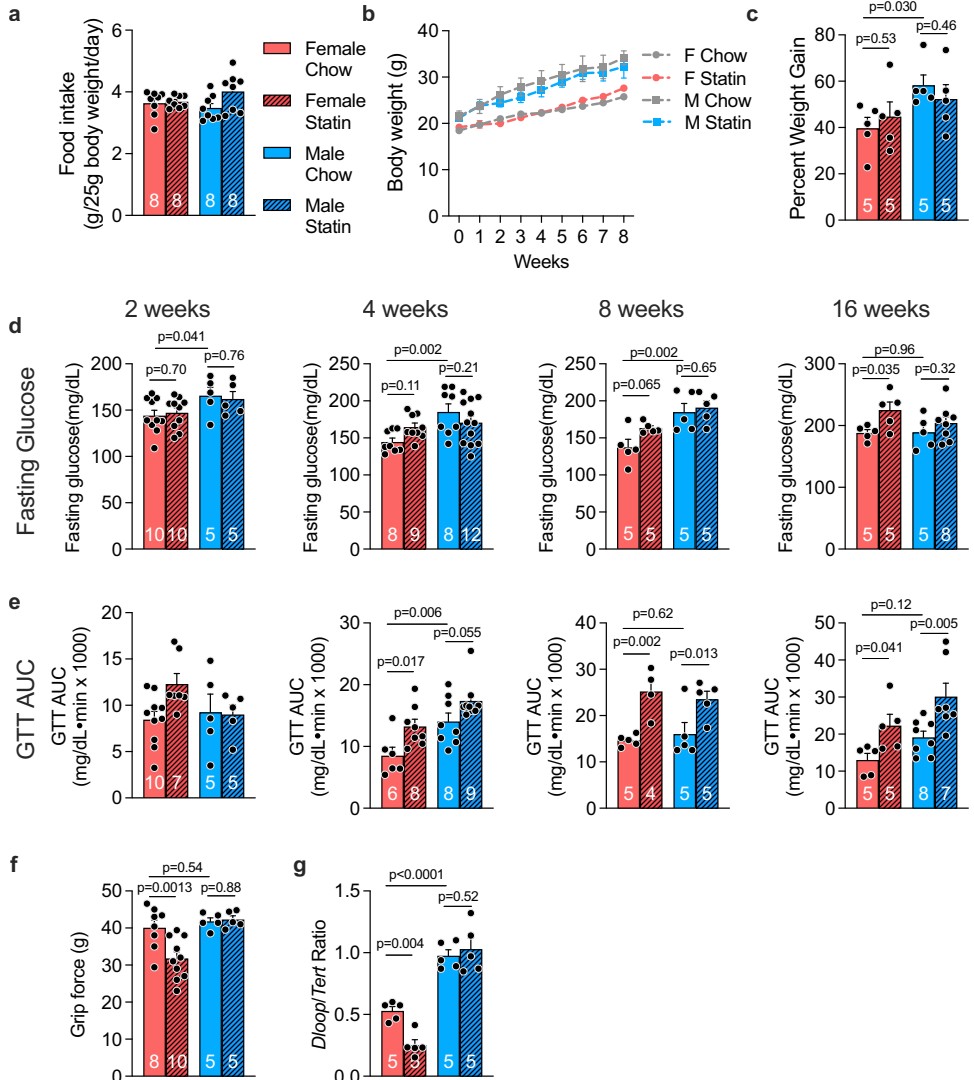

**Fig. 1 | Female sex exacerbates statin-induced changes in glucose homeostasis and grip strength in mouse. a–c** C57BL/6 J male and female mice were fed chow or chow containing simvastatin (0.1 g/kg) for 8 weeks. Daily food intake was assessed for 5 days during week 3 of the diet. Body weight was determined each week for 8 weeks. Percent weight gain (weight gain/body weight) was determined for each mouse at week 8. $N = 8$ independent mice/group for **a**, **b**, and 5 mice/group for **c**. Data are presented as mean ± SEM. **d–e** Independent cohorts of C57BL/6 J mice were fed chow or chow containing simvastatin for 2, 4, 8, or 16 weeks. **d** Fasting glucose and **e** area under the curve during a glucose tolerance test (GTT AUC) were determined for each cohort and are represented as mean ± SEM. The number of

mice/group is indicated on bars in graphs. Data were analyzed by 2-way ANOVA, and where significant, subsequent pair-wise comparisons were performed via unpaired two-sided t-test, with significant comparisons indicated above graphs. **f** Forelimb grip strength after 8 weeks statin treatment in female and male mice, with N for each group shown on bars in graph. Grip force is measured in grams (**g**). **g** Mitochondrial content in quadriceps muscle was assessed after 8 weeks statin treatment for $N = 5$ mice/group. All bars represent mean ± SEM. Data were analyzed by 2-way ANOVA, and where significant, subsequent pair-wise comparisons were performed via unpaired two-sided t-test, with significant comparisons indicated on graphs. Source data are provided as a Source Data file.

tissue glycogen may contribute to sex differences in glucose homeostasis in response to statin.

## Statin treatment impairs the DHA–Nrf2–glutathione axis and mitochondrial activity in female mice

An important role for DHA is to protect against oxidative stress through Nrf2 (nuclear factor erythroid 2-related factor 2) activation and synthesis of glutathione antioxidant (Fig. 4a)[21–24]. The female-specific reduction in DHA levels led us to investigate whether the glutathione axis is compromised in statin-treated females. Glutathione is a tripeptide composed of glutamate, cysteine, and glycine that has a critical role in protecting against environmental toxins, including drugs[25]. DHA and other polyunsaturated fatty acids activate the redox-sensitive Nrf2 transcription factor to promote glutathione synthesis through expression of *Gclc* (encoding the rate-limiting enzyme in

glutathione synthesis), as well as other genes involved in maintenance of cellular redox state (Fig. 4a)[24,26,27].

Consistent with reduced Nrf2 activation, statin-treated females had reduced expression levels of Nrf2 target genes *Sqstm1* (sequestosome 1) and *Pgd* (phosphogluconate dehydrogenase), and trends to reduced *Gsr* (glutathione reductase) and *Txnrd1* (thioredoxin reductase) (Fig. 4b). Compared to males, females also had substantially lower expression levels for the rate-limiting enzyme in glutathione synthesis, *Gclc*, under both basal and statin-treated conditions, but neither sex experienced a reduction in response to statin (Fig. 4c). Gene expression alterations in *Glud1* and *Got1* were consistent with a shift in equilibrium to reduce glutamate levels at the expense of increased α-ketoglutarate in statin-treated females (Fig. 4a, d). Consistent with the gene expression levels, liver and plasma levels of glutamate were reduced in statin-treated females (Fig. 4e), whereas liver levels of α-ketoglutarate were increased

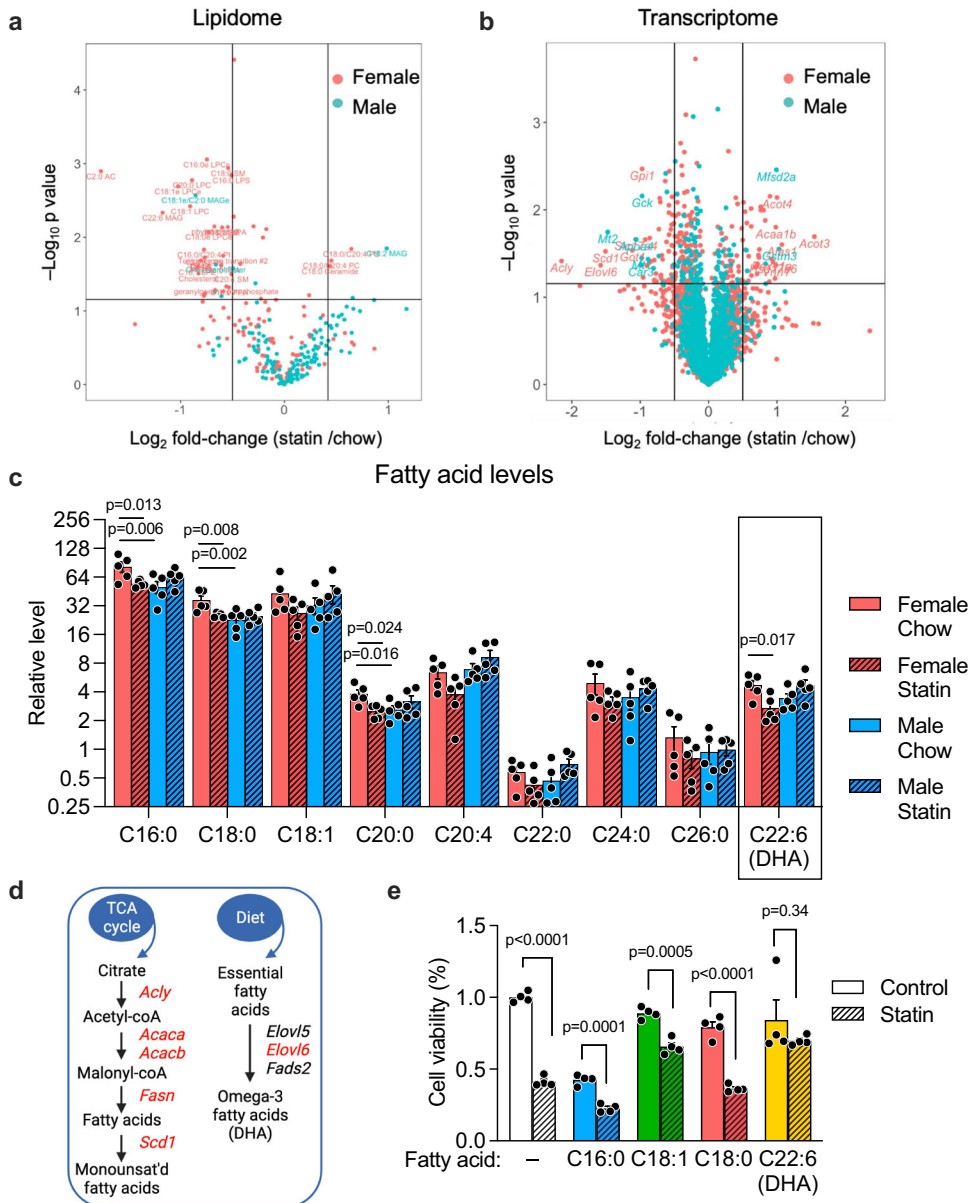

**Fig. 2 | Female-biased alterations in hepatic lipidome and transcriptome after statin treatment. a** Mass spectrometric profiling of liver lipidome in mice fed chow or chow containing statin for 4 weeks (*N* = 5/sex on each treatment). Lipid species with significant differences between statin and control diets within each sex (determined with EdgeR, nominal *p* < 0.05 without correction for multiple testing) are shown above the horizontal line of the volcano plot. The vertical lines delineate a 50% decrease (to the left) or increase (to the right) in levels in response to statin. **b** Volcano plot of RNA-seq gene expression data from liver of mice in **a**. Genes with significant differences between statin and control diets within each sex (nominal *p* < 0.05 without correction for multiple testing) are shown above the horizontal line. The vertical lines delineate 0.5-fold decrease (to the left) or increase (to the right) in mRNA levels in response to statin. **c** Relative levels of fatty acid species from mass spectrometric profiling in female and male mice fed chow without or with statin. Fatty acids are represented as number of carbons followed by number

of double bonds. DHA, docosahexaenoic acid. Statistical analyses were as in Fig. 1 with *N* = 5 mice/sex on each treatment. Values represent mean ± SEM. Significance determined by unpaired two-sided t-test. **d** Pathways for synthesis of monounsaturated (left) and polyunsaturated fatty acids (right). Gene names in red exhibit reduced expression in female mice treated with statin. (Diagram was created with BioRender.com released under a Creative Commons Attribution-NonCommercial-NoDerivs 4.0 International License.) **e** The effect of fatty acids with female-specific reductions in statin-treated mice shown in **c** on hepatocyte viability. Hepa1-6 cells were treated with BSA-conjugated fatty acids ± simvastatin for 24 hr and cell viability determined by spectrophotometric assay as reduction of a tetrazolium salt (MTT). Values represent mean ± SEM. Significance determined by unpaired two-sided t-test with *N* = 4 independent cell wells/treatment without adjustment for multiple testing. Source data are provided as a Source Data file.

(Fig. 4f). Collectively, these findings indicate that statin reduces activation of the DHA–Nrf2–glutathione axis in females.

The reduced form of glutathione (GSH) is a critical antioxidant for maintenance of the mitochondrial redox environment[28]. Statin treatment of female mice led to a decrease in hepatic GSH levels (Suppl. Fig. 5a) and the GSH/GSSG ratio (a measure of reduced to oxidized forms of glutathione) (Fig. 4g). Females also exhibited

reduced mitochondrial complex I, II, and IV activity (Fig. 4Hh compare shaded red to solid red bars). When females received fish oil in combination with statin treatment, the decrements in GSH levels and mitochondrial respiration were largely prevented (Fig. 4g, h, compare shaded red to shaded yellow bars). Male mice did not experience reductions in antioxidant levels nor mitochondrial activity with statin (Fig. 4i, j, Suppl. Fig. 5b), but rather displayed an

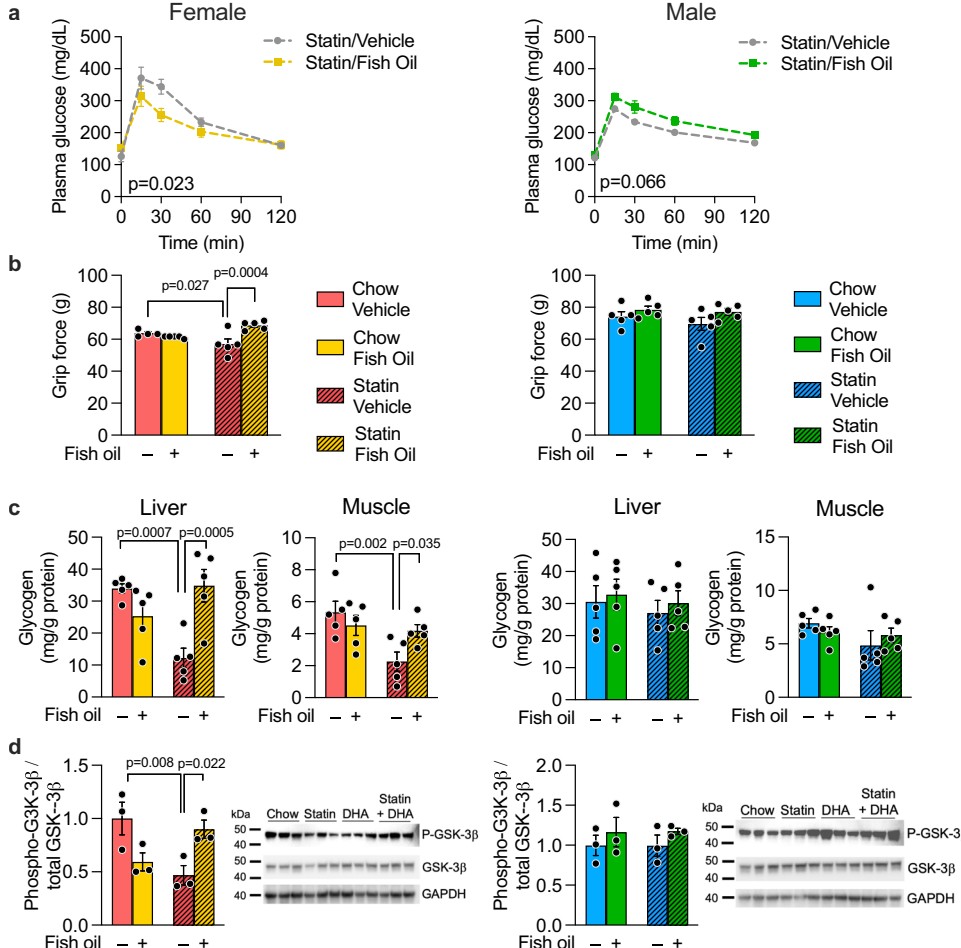

**Fig. 3 | Statin co-therapy with fish oil prevents statin-induced glucose intolerance, impaired grip force, and altered tissue glycogen homeostasis in female mice.** Mice were fed chow or statin-containing diet with daily administration of either coconut oil vehicle or fish oil (as a source of ω-3 fatty acids) dissolved in coconut oil vehicle. After 5 weeks, **a** glucose tolerance and **b** grip strength were measured in female and male mice ($N = 5$ mice/sex on each treatment). AUC, area under the curve for glucose tolerance test. Tissues from mice in **a**, **b** (5 mice/sex on each treatment) were used to determine **c** liver and muscle glycogen levels by biochemical assay, and **d** levels of total and phosphorylated glycogen synthase kinase-3β (GSK−3β) by immunoblot. The same blot was used sequentially for detection of phosphorylated and total GSK−3β and GAPDH. Values represent mean ± SEM. Data analyses were as in Fig. 1 with *p* values representing pair-wise two-sided t-tests following significant 2-way ANOVA. Source data are provided as a Source Data file.

increase in complex I activity on statin, which was normalized by fish oil co-therapy.

## XX chromosomal sex and *Kdm5c* gene dosage increase risk for statin adverse effects

We next investigated what determinants of biological sex influence susceptibility to adverse statin effects in females. Sex differences between males and females can be parsed into genetic sex (XX vs. XY chromosome complement), which is a source of gene dosage differences between males and females, and gonadal sex (ovaries vs. testes), which confers differential gonadal hormone levels in females and males. We used the Four Core Genotypes (FCG) mouse model to interrogate genetic and gonadal sex contributions. This model consists of four sex genotypes−mice with XX chromosomes and either female or male gonads, and mice with XY chromosomes and either female or male gonads[29] (Fig. 5a). Since prescription of statins typically occurs for individuals with hypercholesterolemia, we utilized FCG mice on a C57BL/6 background that develop hypercholesterolemia on a chow diet due to genetic ablation of the apolipoprotein E (apoE) gene. The cholesterol levels for this mouse cohort are 400–600 mg/dL on a chow diet, which are reduced to ~300 mg/dL after statin treatment for 8 weeks[30].

We evaluated statin adverse effects in the apoE-deficient FCG mice fed chow ± statin for 8 weeks. Analysis by 3-way ANOVA (gonadal sex, chromosomal sex, and statin treatment as variables) revealed that sex chromosome complement influences statin-induced impairment in glucose tolerance and grip strength. Specifically, statin treatment in mice with XX chromosomes (regardless of ovaries or testes) led to impaired glucose tolerance (increased values in Fig. 5b, red and teal bars), whereas XY mice were protected (Fig. 5b, purple and blue bars). XX mice also experienced reduced grip strength with statin treatment, whereas XY mice did not (Fig. 5c). Thus, XX chromosome complement is a risk factor for statin adverse effects.

XX chromosome complement may influence metabolism through genes that escape X-chromosome inactivation (X-escape genes) and are therefore expressed at higher levels in XX compared to XY cells[31–33]. We previously identified four X-escape genes with >40% higher expression in XX compared to XY liver[31,34]. One of these X escape genes, *Kdm5c*, encodes a histone demethylase that represses fatty acid biosynthetic gene expression[35], including genes that we found to be reduced by statin treatment in female mice (Fig. 2d). Furthermore, KDM5C histone demethylase activity requires α-ketoglutarate as a cofactor[36], such that the increased α-ketoglutarate levels in statin-treated females (Fig. 4f) may enhance KDM5C repression of gene expression.

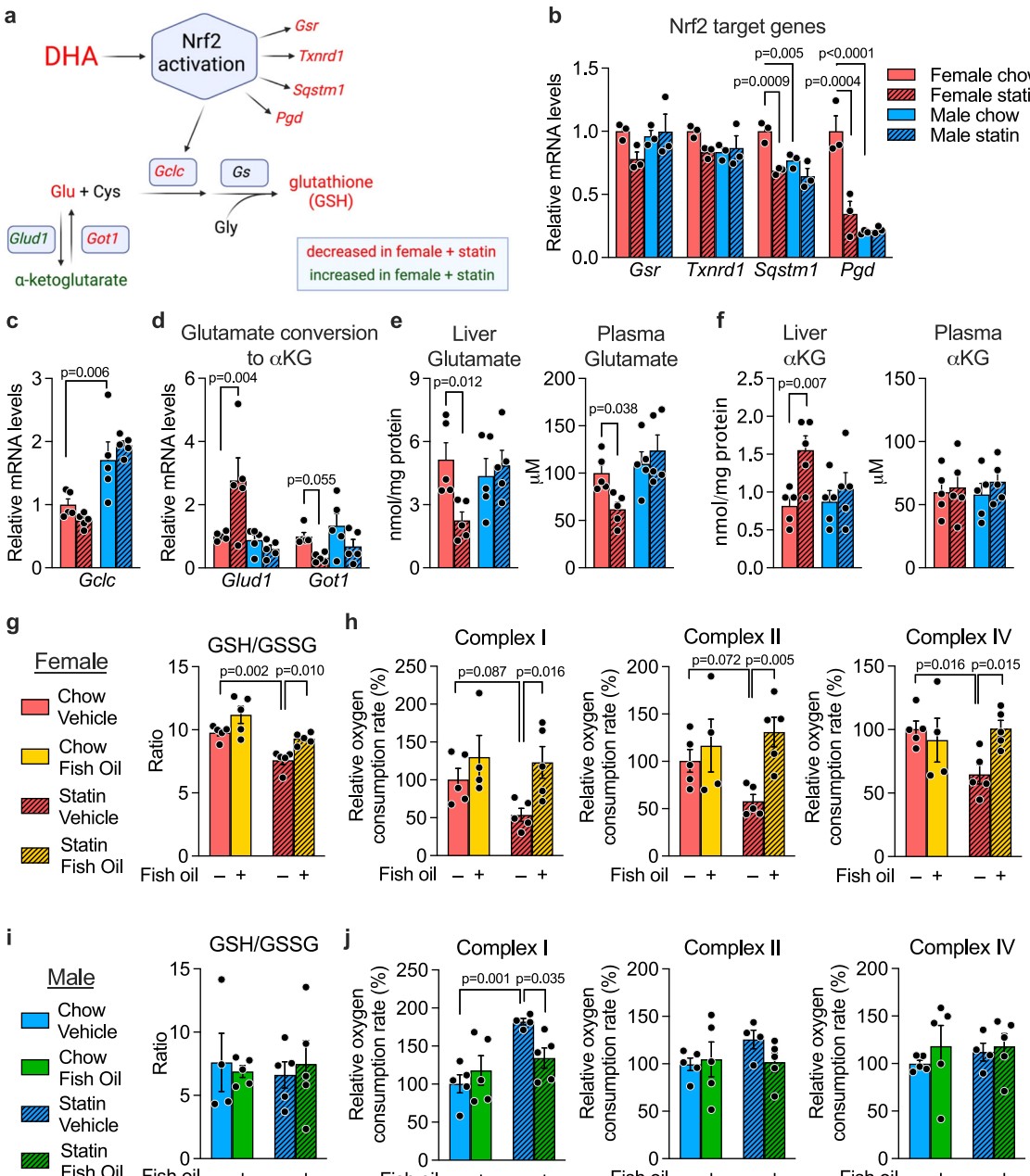

**Fig. 4 | Statin impairs, and fish oil normalizes, glutathione levels and mitochondrial activity in female mice. a** DHA promotes production of the antioxidant, glutathione, through activation of Nrf2 and regulation of its transcriptional targets, including the gene encoding the rate-limiting enzyme in glutathione biosynthesis, *Gclc*. Red text, mRNA and metabolite levels that are decreased in females in response to statin; green text, levels that are increased in females in response to statin; black text, not measured. (Diagram was created with BioRender.com released under a Creative Commons Attribution-NonCommercial-NoDerivs 4.0 International License.) Data for components shown in red and green are provided in subsequent panels of the figure; reduced DHA levels in statin-treated females are documented in Fig. 2c. Hepatic mRNA expression levels for **b** Nrf2 target genes (*Gsr, Txnrd1, Pgd, Sqstm1*) (N = 3 mice/sex on each treatment) and **c** the glutathione synthetic gene, *Gclc* (N = 5 mice/sex on each treatment). **d** Expression levels for *Glud1* and *Got1*, which regulate glutamate (Glu) and α-ketoglutarate interconversion (N = 5 mice/sex on each treatment). **e** Glutamate and **f** α-ketoglutarate levels in liver and plasma (N = 5 mice/sex on each treatment). **g–j** Effect of statin treatment in combination with fish oil or coconut oil vehicle on glutathione levels (ratio of reduced to oxidized forms) in female (**g**) and male mice (**i**), and on activity of mitochondrial complexes I, II, and IV in female (**h**) and male liver (**j**). N = 5 mice/sex on each treatment. All data are represented as mean ± SEM. Significance determined by 2-way ANOVA, and where significant, subsequent pair-wise comparisons were performed via unpaired two-sided t-test. Source data are provided as a Source Data file.

We hypothesized that the higher *Kdm5c* expression levels in XX compared to XY tissues in both mice and humans[31,37,38] may predispose to adverse statin effects on fatty acid biosynthetic gene expression, redox tone, and glucose homeostasis. To test this, we generated female mice with *Kdm5c* gene dosage reduced to a single allele (*Kdm5c^{+/−}*) and determined if they are protected from statin adverse effects compared to females with the normal gene complement (*Kdm5c^{+/+}*) (Fig. 6a). When treated with statin, wild-type *Kdm5c^{+/+}* female mice developed glucose intolerance as we have seen throughout our studies with female mice, but female *Kdm5c^{+/−}* mice were protected from statin-induced glucose intolerance (Fig. 6b, compare red and blue stippled bars). *Kdm5c* gene dosage did not influence grip strength. *Kdm5c^{+/−}* mice also were protected from statin-induced reduction in GSH/GSSG ratio (Fig. 6c) and impaired mitochondrial

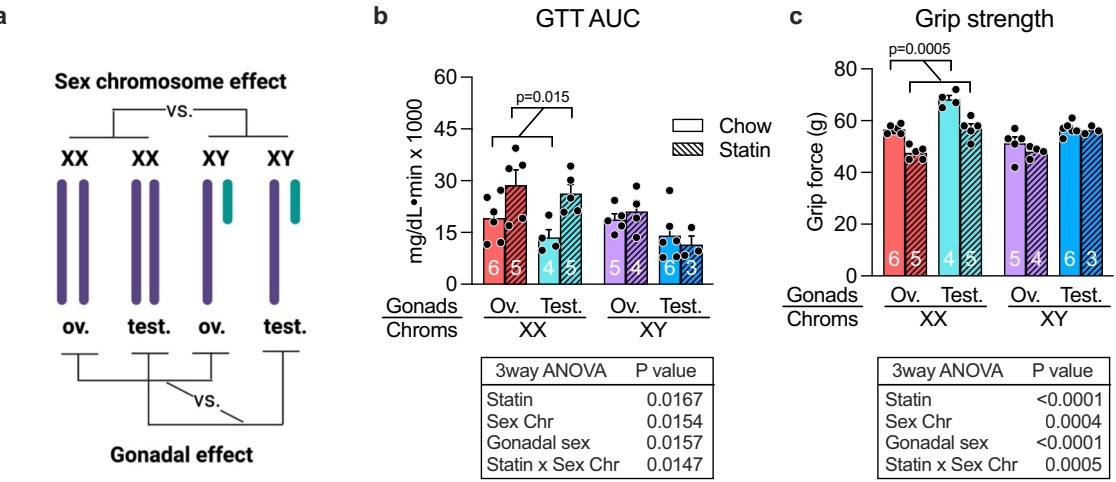

**Fig. 5 | XX chromosome complement segregates with statin adverse effects in mouse. a** The Four Core Genotypes mouse model was used to assess the relative roles of sex chromosomes and gonadal sex in statin adverse effects. The four genotypes of this model are shown with the comparisons that allow assessment of chromosome vs. gonadal effects. Ov. ovaries, Test. testes. (Diagram was created with BioRender.com released under a Creative Commons Attribution-NonCommercial-NoDerivs 4.0 International License.) **b** Glucose tolerance test AUC (area under the curve) after 6 weeks on statin. Number of mice of each genotype and treatment are indicated on bars in graph. **c** Grip strength after 7 weeks on statin. Number of mice of each genotype and treatment are indicated on bars in graph. Values represent mean ± SEM. Results were analyzed by 3-way ANOVA with results shown in tables below graphs. Post-hoc analysis was by pairwise t-tests. Source data are provided as a Source Data file.

complex activity (Fig. 6d), and resisted statin-induced reduction of fatty acid gene expression levels that occur in *Kdm5c*[+/+] mice (Fig. 6e). Thus, reducing female *Kdm5c* gene dosage to a single allele as is present in male mice prevents statin-induced impairment of glucose tolerance, redox tone, mitochondrial activity, and fatty acid gene expression.

## Women are more susceptible than men to statin-induced decrements in DHA ω-3 fatty acid and mitochondrial complex activity

Our studies in mice indicated that sex-biased reduction in DHA levels after statin treatment is a determinant of adverse effects in females, and that fish oil complementation prevents glucose intolerance, reduced oxidative tone, and impaired mitochondrial activity. To determine relevance to humans, we assessed men and women for alterations in ω-3 fatty acid levels (DHA and EPA) in response to short-term statin treatment. Blood was collected from men and women without known cardiovascular disease or type 2 diabetes before statin administration and following 10 weeks of daily statin treatment (atorvastatin, 40 mg)[39] (Fig. 7a). Statin treatment elicited reductions in DHA levels in women and men, but women dropped more than 50 μmol/L compared to 25 μmol/L in men (Fig. 7b). Statin treatment caused a slight decrease in EPA levels in both sexes. Only women showed an increase in glucose levels after 10 weeks of statin treatment (Fig. 7b).

Analysis of all individuals in the study showed a weak but significant correlation between statin-induced decrement in DHA levels and increased glucose levels (Fig. 7c), while no significant correlation was detected between EPA and glucose levels (Fig. 7d). Analysis of only individuals who experienced a decrease in DHA levels in response to statin revealed a stronger correlation between decrement in DHA and increased glucose levels (Fig. 7e), with no relationship between EPA and glucose levels (Fig. 7f).

To determine whether the impaired mitochondrial activity noted in statin-treated female mice is replicated in humans, we utilized patient-derived induced pluripotent stem cells (iPSC) from individuals who were susceptible or resistant to statin-induced new-onset diabetes (NOD) (Fig. 8a). Information was obtained from electronic medical records to identify statin NOD cases and statin-resistant controls. NOD cases had at least two of the following clinical indicators of diabetes arising during statin treatment: elevated fasting glucose levels (>126 mg/dL), diagnosis of diabetes during statin treatment, or first prescription of anti-diabetic drugs during statin treatment. Control subjects experienced none of these indicators of NOD. Prior to statin initiation, NOD cases and controls both had normal fasting glucose measures (<110 mg/dL). iPSCs were derived from CD34+ peripheral blood mononuclear cells by reprogramming with Yamanaka factors; representative iPSC lines were validated for karyotype and expression of pluripotent markers[40].

We treated cultured iPSC lines (*n* = 3 independent lines per sex and disease status) with statin for 24 hr, and performed respirometry on all cell lines simultaneously. We utilized atorvastatin for these studies as we have determined that it produces a robust transcriptional response of genes in the mevalonate pathway in cultured iPSCs at lower concentration than simvastatin. We assessed response of each patient cell line in the presence of statin compared to the absence of statin. Statin treatment impaired complex I, II, and IV activity in cells from women NOD cases compared to controls (Fig. 8b), consistent with our observations in female mice that experience statin adverse effects. This effect was not observed in cell lines from men (Fig. 8c), consistent with our observations in male mice. These results suggest that mechanisms underlying statin-associated NOD in women and men may differ.

As in the mouse, human female tissues express *KDM5C* at higher levels than males[37]. This property is also evident in iPSCs derived from women and men in our study. This is true regardless of statin treatment or NOD status (Fig. 8d). By contrast, the majority of X chromosome genes show similar expression levels in iPSCs derived from women and men (a representative gene, *FLNA*, is shown in Fig. 8d).

## Discussion

Statins remain the most commonly used drug to lower plasma cholesterol levels and reduce the risk of coronary heart disease. Adverse effects experienced by some statin users are a deterrent to continued use, leading to reduced protection from cardiovascular disease. Previous studies have identified genetic variants that influence susceptibility to statin adverse effects in genes encoding the target of statin action (*HMGCR*, HMG-CoA reductase), a statin transporter (*SLCO1B1*

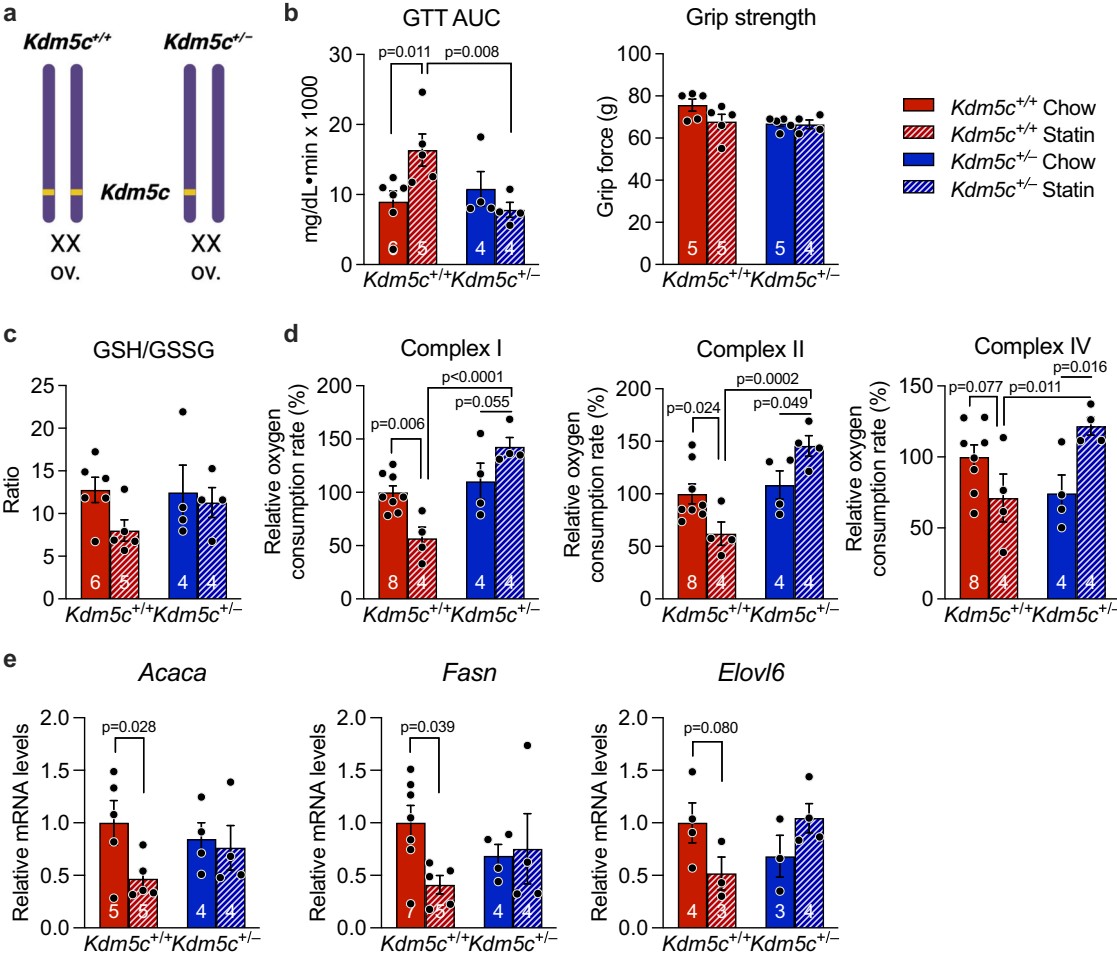

**Fig. 6 | Female mice with *Kdm5c* gene dosage similar to males are protected from statin adverse effects. a** Female XX mice carrying two (*Kdm5c*[+/+], normal female genotype) or one *Kdm5c* gene allele (*Kdm5c*[+/−]) were characterized for statin response. (Diagram was created with BioRender.com released under a Creative Commons Attribution-NonCommercial-NoDerivs 4.0 International License.) After 4 weeks on statin, mice were assessed for **b** glucose tolerance area under the curve and grip strength, **c** ratio of reduced to oxidized glutathione (GSH/GSSG), **d** mitochondrial complex I, II, and IV activities, and **e** fatty acid synthesis and elongase gene expression (see text for full gene names). Values represent mean ± SEM. Data analyzed by 2-way ANOVA followed by pairwise two-sided t-tests (**b**–**d**), or directly compared by pairwise t-test within genotype (**e**). Source data are provided as a Source Data file.

organic anion transporter), and enzymes that metabolize statin for elimination (*ABCB1* transporter and *CYP3A* and *CYP2D6* cytochrome P450 genes)[41–43]. Previous work has also implicated statin-induced reductions in the levels of HMG-CoA pathway lipid intermediates, including coenzyme Q and isoprenoid groups (*e.g.*, farnesol and geranylgeraniol) that covalently modify specific proteins[2,44]. However, these genetic variants and biochemical mechanisms have not been related to differences in statin adverse effects between males and females.

We exploited multiple mouse models and human cohorts to assess sex differences in statin adverse effects and to identify molecular and genetic mechanisms (Suppl. Fig. 6). Our findings indicate that female C57BL/6 J mice have impaired glucose tolerance and muscle grip strength after short-term statin treatment, whereas male mice on the same genetic background resist these adverse effects. Our findings further reveal that female sex interacts with statin treatment to reduce hepatic levels of the ω-3 fatty acid DHA, reduce cellular redox tone, and impair mitochondrial electron transport complex activity. The decrement of DHA as a contributor to sex-biased adverse statin response was supported by the demonstration that adverse physiological and biochemical statin effects are prevented in female mice by supplementation with fish oil as a source of ω-3 fatty acids. Our data indicate that statin adverse effects in female mice are also associated with a reduction

in glutathione levels. It has previously been shown that abnormal glutathione redox status leads to reduced mitochondrial complex I activity[45,46] and to reduced insulin sensitivity[47–49].

At the genetic level, our studies demonstrated that XX sex chromosome complement promotes risk for statin adverse effects compared to XY chromosome complement, regardless of the presence of ovaries or testes. The role of X chromosome dosage as a determinant of sex differences has been shown for conditions such as obesity, autoimmunity, and neurodegeneration[31–33]. The current studies extend the impact of X chromosome dosage to include response to statin. We further determined that female dosage of a specific X chromosome gene, *Kdm5c*, promotes adverse statin effects; we could make female mice resistant by reducing the female *Kdm5c* gene dosage from two (*Kdm5c*[+/+]) to one allele (*Kdm5c*[+/−]). Our previous studies have shown that the tissue levels of KDM5C protein in *Kdm5c*[+/−] mice are about 40% of those in wild-type females[31]. We note that the tissues of *Kdm5c*[+/−] mice contain a mixture of two types of cells: cells in which the nonfunctional allele is on the inactivated X chromosome (and *Kdm5c* expression therefore occurs from a single normal allele), and cells in which the nonfunctional allele is on the active X chromosome (such that *Kdm5c* expression derives from the escape from inactivation on the X). Future studies that examine the transcriptome of *Kdm5c*[+/−] mice at the single cell level will be interesting to identify statin influence on

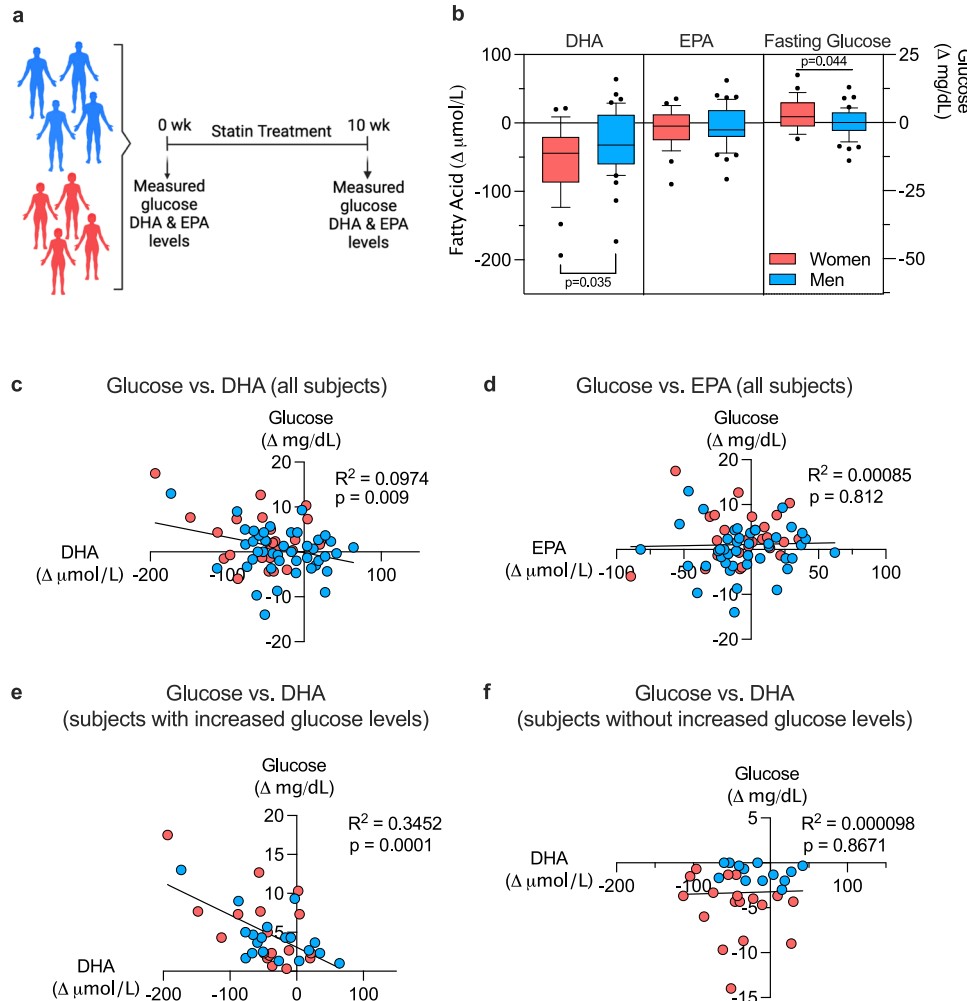

**Fig. 7 | Women are more susceptible than men to statin-induced decrements in DHA levels. a** Blood collected from statin-eligible men and women without type 2 diabetes was used to assess ω-3 fatty acid (DHA and EPA) and glucose levels before and 10 weeks after statin treatment. (Diagram was created with BioRender.com released under a Creative Commons Attribution-NonCommercial-NoDerivs 4.0 International License.) **b** Women showed more pronounced statin-induced reduction in DHA levels, and increased glucose levels, compared to men. Data shown as box plots with median (center line) and 10th and 90th percentile indicated; significance between sexes determined by unpaired two-sided t-test ($N = 26$ women and $N = 43$ men). **c, d** Regression analysis of DHA or EPA levels vs. glucose levels in all subjects from **b** was performed using Prism v10.0. **e, f** Regression analysis of DHA or EPA vs. glucose levels for individuals that reduced fatty acid levels in response to statin treatment ($N = 18$ women and $N = 19$ men) was performed using Prism v10.0. Source data are provided as a Source Data file.

gene expression in individual cells with different levels of *Kdm5c* expression.

The *Kdm5c* gene represents a genetic modifier of statin response, and will be the topic of future investigations. For example, tissue-specific modulation of *Kdm5c* dosage will be useful to zero in on the key tissue(s) that initiate statin NOD. Given the molecular function of KDM5C as a histone demethylase, we hypothesize that sex differences in epigenetic regulation of specific genomic targets contribute to the female-biased effects of statin. Identification of KDM5C-mediated histone modifications that are altered by statin may identify additional genes that influence statin adverse effects.

Our results in the mouse were reinforced by findings that women experience more severe reductions than men in DHA levels after short-term (10 week) statin administration, and that DHA reduction was correlated with increases in fasting glucose levels. The statin-induced reduction in DHA without reduction in the ω-3 fatty acid EPA that we observed is consistent with a previous study in a small Japanese cohort[50]; however, results in that study were not stratified by sex. There is a dearth of studies that include women, so data are limited, but there is evidence that females in particular experience beneficial effects of ω-3 fatty acids on glucose metabolism. For example, ω-3 fatty

acid supplementation improves fasting glucose levels in women with polycystic ovary syndrome, and glucose tolerance in female, but not male, adolescents with obesity[51,52]. Additional translation of our findings in mouse to humans was achieved by demonstrating that iPSCs derived from women who developed NOD exhibited impaired mitochondrial complex activity when treated with statin, as we observed in female mice. We note that studies with patient-derived iPS cells were performed with atorvastatin, whereas in vivo studies were performed with simvastatin. Both statins are known to promote NOD in susceptible individuals; we selected atorvastatin for our studies because it is effective in cultured cells at a lower concentration, most likely because it is administered in its active form, whereas simvastatin is introduced as an inactive lactone derivative.

Our studies with mice and with cultured human cells allow us to assign biological sex distinct from gender as a key component of increased female susceptibility to statin adverse effects. We note that gender may also influence the development and detection of NOD through effects on adherence to statin use, diet, physical activity, and likelihood of seeking medical attention for NOD. Our findings also suggest potential biomarkers for statin adverse effects (reduced DHA levels, glutathione levels, and mitochondrial function) and identify a

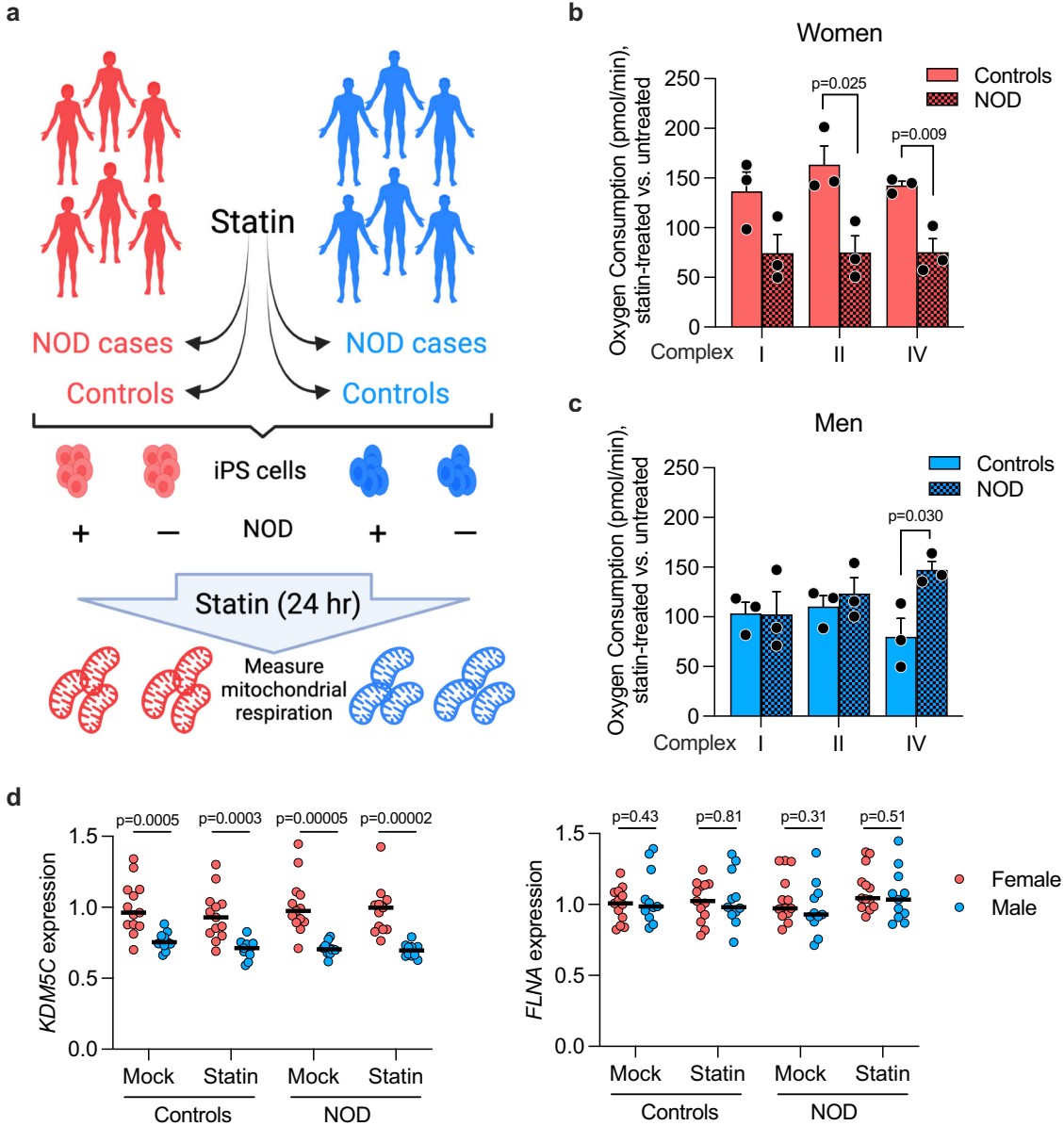

**Fig. 8 | Impaired mitochondrial respiration in patient-derived iPSCs from women with statin new-onset diabetes (NOD). a** iPSCs were programmed from peripheral blood mononuclear cells obtained from men and women with documented diabetes development (NOD cases) or resistance (controls) during statin use. iPSCs were cultured ± statin for 24 hr and frozen. (Diagram was created with BioRender.com released under a Creative Commons Attribution-NonCommercial-NoDerivs 4.0 International License.) Mitochondrial respirometry was performed on all samples in parallel using methodology for frozen samples (see Methods). Mitochondrial complex I, II, and IV respiration in NOD case and control **b** women and **c** men. Values represent mean ± SEM. Significance determined by unpaired two-sided t-test between control and NOD individuals with $N = 3$/sex in each disease status. **d** Mean gene expression levels in iPSCs from men and women control and NOD subjects treated ± statin for 24 hr. *KDM5C* is an X-escape gene that is expressed at higher levels in females compared to males. *FLNA* is an X chromosome gene that undergoes silencing due to X inactivation and is expressed at similar levels in females and males. Significance determined by unpaired two-sided t-test between sexes within disease status and treatment (not adjusted for multiple comparisons). $N = 13$ women and $N = 11$ men. Source data are provided as a Source Data file.

candidate gene (*KDM5C*) whose expression levels influence development of adverse effects. Finally, our findings suggest a major role for altered ω-3 fatty acid metabolism in the pathogenesis of statin-induced NOD and myopathy, and raise the possibility that supplementation with DHA may be of value in ameliorating or preventing these adverse effects.

## Methods
### Ethics statement
Animal studies were performed under approval from the UCLA Institutional Animal Care and Use Committee (protocol ARC 2006-161). Studies of human blood samples were performed with approval of the Stanford University Institutional Review Board (IRB 33347). Human iPSCs were generated under approval of the Kaiser Permanente of Northern California Institutional Review Board (IRBNet 1277833-16) and UCSF Benioff Children's Hospital Oakland Institutional Review Board (IRB #2015-018). Individuals of both sexes were recruited for all studies.

### Mouse strains
Male and female C57BL/6 J mice were obtained from The Jackson Laboratory (Bar Harbor, ME). C57BL/6 Four Core Genotypes mice with *Apoe* deficiency were described previously[53] and were bred in-house as described for standard Four Core Genotypes mice. *Kdm5c*$^{+/+}$ and

$Kdm5c^{+/-}$ were on a C57BL/6 background (floxed mice were originally backcrossed to C57BL/6 J for 11 generations and then crossed to E2a-Cre transgenic mice)[31]. Male mice with global hemizygous $Kdm5c$ deficiency ($Kdm5c^{-/Y}$) were not recovered and thus only female mice with reduced $Kdm5c$ gene dosage were studied. Genotyping primers for Four Core Genotypes $Apoe^{-/-}$ and $Kdm5c$ mice are provided in Supplementary Table 1. Mice were maintained on a 12 h light/dark cycle with an ambient temperature of 21–22 °C and humidity of 50–60%. All animal studies were approved by the UCLA Institutional Animal Care and Use Committee.

## Administration of statin and fish oil to mice

From weaning, all mice were maintained on standard mouse chow provided by our institutional Department of Laboratory Medicine. At 8–10 weeks, mouse cohorts were divided into animals receiving chow diet alone (diet D1001 containing 10 kcal% fat, 20 kcal% protein and 70 kcal% carbohydrate; Research Diets, New Brunswick, NJ) or the identical chow containing simvastatin (0.1 g/Kg food weight in mouse chow, formulation D11060903i, Research Diets) similar to previous reports[54–58]. A simvastatin dose of 0.1 g/Kg in mouse is similar to 80 mg/day in humans based on the FDA conversion factor of 12.3 for "human equivalent dose" in a mouse. Thus, 80 mg per 60 Kg human = 1.3 mg/Kg. A dose of 0.1 mg/g (i.e., 0.1 g/Kg) for a 25 g mouse that eats 4 grams per day = 0.4 mg/25 g body weight = 16 mg/Kg. Finally, 16 mg/Kg/12.3 (FDA conversion factor) = 1.3 mg/Kg.

For studies of ω-3 fatty acid co-therapy, fish oil dose was determined by metabolic rate adjustment of a human high-dose ω-3 fatty acid regimen[59,60]. Mice fed chow or statin diets for 5 weeks were gavaged with a 10 μL/g bolus of coconut oil vehicle (Natural Source, New York, NY) or mixture of fish oil (Nordic Naturals, Watsonville, CA) and coconut oil (1:4), resulting in supply of 250 mg/Kg of DHA and 372.5 mg/Kg of EPA. Oral gavage was performed with 20 gauge plastic feeding tubes (FTP-20-30, Instech Laboratories, Plymouth Meeting, PA) 5 days/week.

## Glucose tolerance and grip strength measurements

Mice were fasted 5 hr (0800–1300) and a blood sample collected to determine fasting glucose levels. To assess glucose tolerance, mice were then injected intraperitoneally with glucose (2 g/Kg body weight) and blood samples collected via tail nick at 30, 60, 120, and 180 min. Glucose determinations were made with an AlphaTRAK glucometer (Zoetis, Parsippany, NJ). Glucose tolerance area under the curve (AUC) was determined using trapezoidal analysis of blood glucose values as described[61]. Plasma insulin levels were determined with a Mouse Ultrasensitive Insulin ELISA kit (ALPCO, Salem, NH). HOMA-IR was calculated as described[31].

Grip strength was measured in studies in Fig. 1 by spring-scale dynamometer (model 8261-M, Ohaus, Parsippany, NJ). Subsequent studies in Figs. 3, 5, 6 were performed with a grip strength apparatus that we constructed with an Arduino microcontroller that interfaces with OSX/Windows operating system[62]. The two methods gave slightly different absolute force measurements, but data obtained within each experiment are internally comparable.

## Mitochondrial DNA quantitation, RNA-sequencing, and qPCR

RNA was isolated from snap-frozen liver samples with TRIzol (Thermo Fisher Scientific). Reverse transcription was performed with iScript and qPCR with SsoAdvanced SYBR Green Supermix (Bio-Rad, Hercules, CA) using primers in Supplementary Table 1 and Bio-Rad CFX Maestro v1.1 software. For mitochondrial DNA quantitation, cellular DNA was isolated by phenol/chloroform extraction followed by ethanol precipitation to recover both genomic and mitochondrial DNA. Mitochondrial DNA was quantitated by qPCR of the mitochondrial D-Loop region (primers aatctaccatcctccgtgaaacc and tcagtttagctaccccaag tttaa) and normalized to genomic DNA, which was quantitated by

qPCR of the $Tert$ gene (primers ctagctcatgtgtcaagaccctctt and gccagcacgtttctctcgtt)[12].

RNA-seq was performed as described previously[30] on $n = 3$ livers from female and male C57BL/6 mice each fed chow or chow-statin diets for 4 weeks. RNA-seq libraries were generated as we have previously[31] with a workflow consisting of poly (A) RNA selection, RNA fragmentation, oligo(dT) priming and cDNA synthesis, adaptor ligation to double-stranded DNA, strand selection and PCR amplification to generate final libraries. Following quantitation (Qubit) and quality-check (4200 TapeStation, Agilent), index adaptors were used to multiplex samples. Sequencing was performed at the UCLA Technology Center for Genomics & Bioinformatics on a NovaSeq 600 sequencer to obtain 50 bp paired-end reads. The average read depth was $18 \times 10^6$ reads/sample, with 85% of reads aligning uniquely to the genome (GRCm38.97). Differential expression was identified by EdgeR v. 3.28.1 (adjusted $p < 0.05$)[63]. Pathway analysis for differentially expressed genes was performed with Enrichr[64]. Volcano plots were generated with ggplot2 v.3.3.6.

## Metabolomic analyses

Nonpolar lipid metabolites were quantified in mouse liver, blood, and quadriceps muscle extracts as previously[65]. Livers were extracted in chloroform:methanol:PBS with inclusion of internal standards C12:0 dodecylglycerol (10 nmol) and penta-decanoic acid (10 nmol). Organic and aqueous layers were separated, and the aqueous layer was acidified [for detection of metabolites such as lysophosphatidic acid (LPA) and LPA-ether (LPAe)], followed by re-extraction with chloroform. The organic layers were combined, dried down, and analyzed by both single-reaction monitoring (SRM)-based LC-MS/MS as described[65]. Metabolites were quantified by integrating the area under the peak and were normalized to internal standard values and then levels from statin-treated animals were expressed as relative levels compared with control animals.

## Fatty acid treatment and cell viability assays

Hepa1-6 mouse hepatocytes (ATCC CRL-1830) were cultured in Dulbecco's Modified Eagle's medium supplemented with 10% fetal bovine serum until they reached 70% confluence. Cells were then treated with fatty acids conjugated with fatty acid-poor bovine serum albumin (BSA, Fraction V, #10775835001, Millipore sigma). Fatty acid treatment was with 100 μM palmitate (C16:0; #P0500, Millipore Sigma), oleate (C18:1; ##O1383), stearate (16:0; #175366), or 2 μM DHA (C22:6; #PHR1788-1G) in the absence or presence of 10 μM simvastatin. After 24 hr, cell viability was assessed using a colorimetric MTT assay kit (#CT02, Millipore Sigma) as directed by the manufacturer.

## Biochemical measurements

Plasma glutamate, α-ketoglutarate, and glutathione levels were assayed using enzymatic kits (Glutamate Assay Kit, MAK004-1KT, Millipore Sigma; α-Ketoglutarate Assay Kit, MAK054-1KT, Millipore Sigma; Glutathione Assay Kit, #703002, Cayman Chemical) according to the manufacturer's instructions. Prior to glutamate or α-ketoglutarate assays, plasma was deproteinated using a Microcon-10kDa centrifugal filter unit (#MRCPRT010, Millipore Sigma). For glutathione (GSH) assays, plasma was deproteinated using MPA reagent (metaphosphoric acid, 1 g/ml; #239275, Millipore Sigma) and TEAM reagent (triethanolamine, #T58300, Millipore Sigma). Direct assessment of deproteinated samples provided reduced glutathione (GSH) levels. To assay levels of the oxidized form of glutathione (GSSG), GSH in deproteinated samples was derivatized with 2-vinylpyridine (#13229-2, Millipore Sigma) prior to assessment. To assay glutamate, α-ketoglutarate, and glutathione in liver, homogenates were prepared in ice-cold buffer specific for the component being analyzed. Buffer was provided in the Glutamate Assay Kit (for glutamate assessment), in the α-ketoglutarate assay kit (for α-KG

assessment), or was 50 mM 2-(N-morpholino)ethanesulfonic acid (MES) buffer (pH6) with 1 mM EDTA (for glutathione assays)[66–68]. Samples were deproteinated and assayed as described above for plasma samples.

Glycogen levels were determined from 50 mg pieces of mouse liver or quadriceps muscle. Extracts were prepared by homogenizing tissue in a Dounce homogenizer in the presence of protease inhibitors (#1183617001, Millipore Sigma), followed by centrifugation at 88 × g for 10 min at 4 °C. Protein concentration in the supernatant was determined using the Pierce BCA kit (#23227, ThermoFisher Scientific). Protein samples were diluted 1:20 for liver and 1:3 for muscle and glycogen determined as specified by the manufacturer using a kit from Cayman Chemicals (#700480).

Total and phosphorylated glycogen synthase kinase (GSK) 3β protein levels were determined by immunoblot using antibodies (1:1000 dilution) that recognize phospho-GSK-3β Ser9 (Cell Signaling Technology rabbit polyclonal antibody #9336, lot 14) or total GSK-3β protein (Cell Signaling Technology monoclonal antibody #9315, clone 27C10). GAPDH protein was detected as a loading control using antibody from GeneTex (#GTX100118, rabbit polyclonal at 1:5000 dilution).

### Mitochondrial respiration assays

Mitochondrial respirometry was performed using homogenates of frozen liver with cytochrome C reconstitution as described[69,70]. This method allows the assessment of complex I, II, and IV activities. Briefly, homogenates were prepared in MAS buffer (70 mM sucrose, 220 mM mannitol, 5 mM $KH_2PO_4$, 5 mM $MgCl_2$, 1 mM EGTA, 2 mM HEPES, pH7.4) in a Dounce homogenizer, cleared by centrifugation (1000 g for 10 min, 4 °C), and the final supernatant was collected and protein quantified with Bio-Rad Protein Assay (Hercules, CA). Respirometry studies to detect Complex I, Complex II, and Complex IV activity were performed with a Seahorse XF96 analyzer (Agilent Technologies, Santa Clara, CA) in MAS buffer containing 10 μg/ml cytochrome C. Substrate injections were: Port A)−pyruvate + malate (5 mM each) for Complex I, 5 mM succinate (Complex II), or 5 mM rotenone (Complex IV); Port B− 2 μM rotenone + 4 μM antimycin; Port C−0.5 mM TMPD (tetramethyl phenylene diamine) + 1 mM ascorbic acid; and Port D−50 mM azide. Data were collected and analyzed with Seahorse XF96 Analyzer v1.4.2 (Agilent).

### Human blood samples

Samples assessed here were collected previously[39]. Written informed consent obtained for these samples allowed their use in subsequent studies; studies were approved by the Stanford University Institutional Review Board (IRB 33347). Briefly, volunteers who were eligible for statin therapy for cardiovascular disease prevention and did not have type 2 diabetes, statin intolerance, or other exclusion criteria [detailed in ref. 39] provided written informed consent. Data from these subjects was previously published demonstrating that atorvastatin treatment for 10 weeks increased insulin resistance in individuals without type 2 diabetes[39], but no assessments of ω-3 fatty acids nor sex stratification were performed in the previous study. For data presented here, ω-3 fatty acids DHA and EPA were assessed in blood samples obtained at week 0 and at week 10 of atorvastatin treatment in 70 subjects (44 men and 26 women). DHA and EPA levels were determined using the Serum Comprehensive Fatty Acid Panel (C8–C26), which employs gas chromatography/tandem mass spectrometry (Quest Diagnostic Nichols Institute, San Juan Capistrano, CA). Glucose levels in these samples were previously determined[39].

### Patient-derived iPSCs

We used electronic health records from Kaiser Permanente of Northern California (KPNC) to identify statin users who developed new-onset diabetes (NOD) and controls who did not develop NOD (Supplementary Table 2). NOD cases were defined as those who had their first statin (simvastatin, lovastatin, atorvastatin, pravastatin, rosuvastatin, or pitavastatin) prescription at age 40–75, and documented continuous statin use for 3 years (Supplementary Data 2). Continuous statin use was defined as having greater than eight 30-day prescription refills per year or greater than three 90-day prescription refills per year. Individuals prescribed statin combinations were excluded.

Individuals with evidence of diabetes prior to the date of the first statin prescription or within the first 3 months after the start of statin use were excluded (Supplementary Data 2). Evidence of diabetes included either (i) diabetes diagnosis based on ICD9 codes for type 1, 2 or gestational diabetes or (ii) prescription for glucose lowering drugs (alpha-glucosidase inhibitors, dipeptidyl peptidase-4 inhibitors, meglitinides, sulfonylureas, biguanides, and thiazolidinediones) and insulin. In addition, individuals with prescriptions for glucose raising drugs, such as oral corticosteroids or with ICD9/CPT codes for bariatric surgery, in the 3 years prior to start of statin treatment, through the 3 years after the start of statin treatment were excluded.

To ensure that subjects were not diabetic prior to the start of statin treatment, they were required to have at least two fasting glucose (FG) measures of 50–110 mg/dL and within 30 mg/dL of each other. Subjects with an outpatient FG measure ≥ 126 mg/dL within the first 3 months after the start of statin treatment were excluded. FG values < 30 mg/dL or >600 mg/dL were also excluded. Subjects who developed NOD in the 3 years after statin initiation were identified and recruited, with diabetes defined as two or more of the following diagnostic elements: (1) FG ≥ 126 mg/dL, (2) diabetes ICD9 code, and/or (3) prescription of a glucose lowering drug (Supplementary Data 2).

Controls were defined as statin users with normal glycemia (all FG values of 50–110 mg/dL) in the 3 years prior to statin initiation and during the first 5 years on-treatment with evidence of continuous statin prescription. Individuals with a diabetes ICD code or use of glucose-modifying drug during the 8 year window were excluded. Blood samples were drawn from identified NOD cases and controls, with clinical and demographic information shown in Supplementary Data 2. Written informed consent was obtained from all study subjects and studies were performed with approval from the Institutional Review Boards at Kaiser Permanente Northern California and the UCSF Benioff Children's Hospitals.

Patient-derived iPSCs were prepared from the participants (Supplementary Data 2). Briefly, iPSCs were reprogrammed from CD34[+] peripheral blood mononuclear cells (PBMCs), and authenticated as previously described[40]. Cells were cultured in mTESR1 media at 37 °C at 5% $CO_2$ and passaged using accutase (Stemcell Technologies, Cat. # 07920) and media supplemented with Y-27632 2HCl (ROCK) inhibitor (Selleckchem, Cat. # S1049; 10 μM working stock). Cells were incubated with either 250 μM atorvastatin (Sigma-Aldrich #189291) dissolved in ethanol or mock buffer (ethanol only) for 24 h, after which cells were washed with PBS and flash frozen. Mitochondrial respiration was determined as described[69,70] and detailed under "Mitochondrial respiration assays."

### Statistics

Statistical tests were performed using Prism v10.0 (GraphPad Software, San Diego, CA) and are indicated in figure legends. In general, in studies with four groups (as in Fig. 1), data were first analyzed by 2-way ANOVA. For analysis of studies with Four Core Genotypes mice on control and statin diets for a total of 8 groups (Fig. 5), groups were compared with three-factor ANOVA with main factors of sex chromosome complement (XX vs. XY), gonadal sex (ovaries vs. testes), and treatment (chow vs. statin). When two-way or three-way ANOVA analyses were significant ($p < 0.05$), subsequent relevant pairwise analyses were performed with unpaired two-sided Student's $t$ test, and results of these pairwise comparisons are shown as $p$ values in Figures. Error bars represent SEM. Statistical analysis of RNA-seq data is described in the Methods section.

## Reporting summary

Further information on research design is available in the Nature Portfolio Reporting Summary linked to this article.

## Data availability

Source data are provided with this paper for Figs. 1–8 and Suppl. Figs. 1–5. The RNA-seq data associated with Fig. 2 have been deposited in GEO (GSE184588). Lipidomics data associated with Fig. 2 have been deposited in MetaboLights (MTBLS9677) and are also provided in Supplementary Table 1. Any additional information is available upon request to the corresponding author (Karen Reue, reuek@ucla.edu). Source data are provided with this paper.

## Materials availability

We will provide mouse strains described in this study upon request to noncommercial entities for research purposes in agreement with University of California regulations. The iPSC lines studied here are available upon request for academic collaboration.

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

## Acknowledgements

We thank the POST study subjects, as well as Brendan Neilan and Gilbert Nalula, for their assistance in the generation of the iPSC lines. These studies were supported by P50 GM115318 from the National Institute of General Medical Sciences (R.M.K., M.W.M., K.R.), U54 DK120342 from the National Institute of Diabetes and Digestive and Kidney Disease/Office of Research on Women's Health (K.R.), U54 HL170326 from the National Heart Lung and Blood Institute/Office of Research on Women's Health, R21 AR077782 (K.R. and P.Z.), R01 DK128898 (K.R. and P.Z.). The Iris Cantor-UCLA Women's Health Center/National Center for Advancing Translational Sciences and UCLA Clinical and Translational Science Institute (K.R.), NIH pre-doctoral fellowships F31 F31AA028183 (J.J.M.) and T32DA024635 (J.J.M.), American Heart Association post-doctoral fellowship 20POST35100000 (C.B.W.), and Doris Duke Foundation, R01 DK116750, P30 DK116074, and R01 DK120565 (J.W.K.).

## Author contributions

P.Z., J.J.M., and K.R. designed studies and wrote the manuscript. P.Z., J.J.M., E.R., and K.S. performed mouse studies and/or analyzed mouse tissues. P.Z. analyzed lipodome data, and C.B.W. analyzed transcriptome data and human fatty acid data and prepared figures and statistical analyses. L.V. performed respirometry studies. J.C.L. generated mouse cohorts and aligned RNA-seq data. D.K.N. provided lipidomic measurements. C.I., M.L., G.S., and A.O.-O. analyzed electronic medical records and identified NOD and control subjects. E.T. assembled tables related to human patient-derived iPSC lines and determined their gene

expression levels. F.A. and J.W.K. provided human blood samples for ω-3 fatty acid analyses; M.J.M. performed ω-3 fatty analyses. M.W.M., A.M., and Y.L.K. derived and cultured iPSCs. M.W.M. and R.K. advised on statin studies and edited the manuscript. All authors reviewed the manuscript.

## Competing interests

The authors declare no competing interests.
