## [Peer Review File · Nature Communications]

X chromosome dosage drives statin-induced dysglycemia and mitochondrial dysfunctionREVIEWER COMMENTS

Reviewer #1 (Remarks to the Author):

In this manuscript, Zhang et al. report findings supporting previous observations suggesting women are more likely than men to experience statin-induced NOD and muscle related side effects. In a series of in vivo experiments and mechanistic studies, the authors show that short-term treatment with simvastatin resulted in worsening of glucose tolerance, increased fasting glucose and muscle weakness in female mice. Follow up studies identified that reduced levels of hepatic DHA, and impaired fatty acid metabolism, mitochondrial respiration, and redox balance, could be important contributing factors. This mechanism is further supported by analyses of gonadal sex and chromosomal sex differences that elucidate molecular and genetic determinants. When taken together, their findings suggest the potential for DHA supplementation to prevent statin-related side effects in woman.

The authors should be commended for investigating this important issue and for their efforts to uncover important sex differences in drug responses. While the findings linking statin-related side effects to altered lipid metabolism, mitochondrial function, and DHA are somewhat novel and interesting, there are several concerns related to the experimental approach and statistical analyses that affect the overall strength of the manuscript. The manuscript would also significantly benefit from additional mechanistic investigations that provide insight into the sex-specific actions of DHA in liver and how this is connected to skeletal muscle function.

Major concerns:

1. Statin dosing and exposure in in vivo models. In providing the rationale for in vivo model dosing, the authors reference two papers, Muraki et al., 2012 and Yokoyama et al., 2007. The studies of Muraki et al focus on atorva and prava, so it would appear the methodological rationale comes primarily from Yokoyama et al. Here the studies were conducted in a different genetic model in male mice only. Given the importance that a clinically relevant dose be studied, and perhaps more importantly, that there are not differences in drug exposure between sexes, the authors should demonstrate that dosing produces simva exposure equivalent to humans in their models and that no differences are observed between male and females. This is critical to interpret findings in the model and ensure that the observed effects do not arise from differences in PK. It would also be interesting to know whether statin liver exposures are different between sexes.
2. In figure 1, there appears to be differences in "chow" fasting glucose levels between male and female mice. Are they statistically different? This would be important to know when interpreting drug effects. Same for lipidomics, are there sex differences between "chow" treatment group?
3. More data are required to facilitate interpretation of results. Are there sex differences in body weight gain/body composition, food intake, etc.
4. Given that multiple tissues (including skeletal muscle, adipose, pancreas, etc) have been implicated in statin-induced NOS, it is not clear why the authors focused on liver lipid metabolism in the in vivo model. The impaired GTT could result from effects in one or more of these tissues. Evidence supporting a liver-specific mechanism is required as is a more thorough investigation into the other tissues.
5. A deeper investigation into skeletal muscle is required to interpret the grip strength data. It is not clear how this fits into the working hypothesis.
6. The changes in hepatic fatty acid metabolism genes and metabolites (SF2 and F2C) appear to be

robust and widespread.

a. Please provide a rationale for focusing specifically on DHA

b. Are there differences in liver, muscle, and plasma TG mass? Sex differences?

7. Please provide rationale for using atorva vs simva for iPSC experiments and why tested at 250uM, a conc. orders of magnitude above free drug levels in humans

Minor concerns:

1. Please report AUC data for figures 1a&b and 2e&f- how they are calculated and statistical methods

2. Please explain discordance between grip strength results for figures 1 and 2. Did statins impair grip strength in F2e? If not, what is the interpretation of the fish oil results.

Reviewer #2 (Remarks to the Author):

The manuscript entitled "X chromosome dosage drives statin-induced dysglycemia and mitochondrial dysfunction" by Zhang and colleagues, brings some experimental evidences in mice and humans, to explain why woman are more susceptible to the side effects on muscle and glucose homeostasis induced by statin treatment.

Please find below my main comments:

- Literature about gender and myopathy is not really solid and the Authors only cited one paper that support this idea. This cause-effect relationship doesn't seem to be a stablished concept.

- The Authors should have add a group of hypercholesterolemic mice treated with statins to try to reproduce what happens in humans who are treated with statins.

- The poor metabolic phenotyping in Figure 1 is an evident shortcoming of this study. The Authors do not provide data on insulin resistance, although some degree of high glucose concomitant with blood insulin indicates the females might be insulin resistant. Does that match with the literature?

- GTT experiments reflects changes in glucose uptake too, but mainly reflects the glucose induced insulin secretion. There is no dosage of insulin along the glucose dosages that may explain the results.

- Elevated fasting glucose may reflect an enhanced hepatic glucose production which was not evaluated here. Many other experiments should have been provided, such as gene/protein expression of gluconeogenic genes and proteins, pyruvate test, etc. Moreover, the GTT experiments at Fig.1 do not display the AUC or statistical analysis, albeit it seems that the intolerance was observed in both genders. Lack of statistical analysis and AUC for the GTTs makes the interpretation of the data a little difficult.

- It is not clear in text how the Fatty acids quantified in the Figure 2C are in the free form or it is a compilation of the FAs incorporated in the different forms (i.e. as LPC, CL, PC, TAGs, DAGs, etc...). A more detailed presentation of the lipidomic data should be provided.

- It seems that all the long-chain FAs showed in Fig 1C, including saturated and monosaturated ones,

are reduced. The lack of this pool of FAs can lead to the malfunction of mitochondrial machinery, since this results in a lack of FA substrates for fatty acid oxidation pathway in mitochondria and consequent lack of further generation of intermediates for the OXPHOS at electron transport chain.

- The Authors did not provide the composition of the fish or coconut oil, although it only indicates the total amount of DHA and EPA included. However, several other FAs are being provided together with coconut and fish oil, and it is important to know which ones and how much of them. Moreover, it doesn't seem that the circulatory levels of DHA and EPA were provided after fish oil treatment. Did the treatment restore the DHA levels in the liver?

- The link between the mitochondrial alterations and the dysglycemia is not clear, since there is no mechanistic demonstration that clarifies this hypothesis.

- CoQ10 is a key component of the mitochondrial electron transport system. Therefore, CoQ10 deficiency resulting from statin treatment may impair cellular energy metabolism, and contribute to the development of myopathy and muscle symptoms, as described in patients treated with statins in many previous studies (<https://doi.org/10.1016/j.jacc.2007.02.049>, doi: 10.1023/B:CARD.0000015861.26111.ab., doi: 10.1001/jama.289.13.1681., doi: 10.1002/biof.5520180221). It is very likely that CoQ10 deficiency is the key link between statin treatment and the results observed in the females, CoQ10 should be also measured in the other conditions including the knockout mice and fish-oil treated mice. There are evidences that DHA may increase CoQ10 (doi: 10.1111/mcn.12300 doi.org/10.1016/j.redox.2021.102061).

Reviewer #3 (Remarks to the Author):

The study by Zhang et al. explores the potential mechanisms by which women experience adverse effects to statin therapy relative to men. The experiments provide a number of interesting findings, including that female mice on simvastatin therapy are characterized by glucose intolerance, fasting hyperglycemia, lower grip strength, decreased DHA levels, decreased hepatic GSH/GSSG ratio, and decreased electron transport complex I, II and IV enzyme activities. Interestingly, supplementing statin-treated female mice with fish-oil prevents the adverse effects. The data is presented logically and the manuscript well-written.

The main concern with the study is that the effect of statin and statin counter therapies were evaluated in lean chow fed mice, not in the context of obesity, western diets, and/or hypercholesterolemia. It is not clear why potential differences in statin adverse effects between sexes were not studied under more translationally relevant conditions. While the argument can be made that chow fed mice represent a more controlled condition, it is entirely possible that the sex differences identified are specific to the context in which they were studied and therefore of limited translational relevance.

Other areas for potential improvement include the following:

Assessment of muscle function/structure/myopathy limited to grip strength. The study could be

strengthened considerably by analysis of skeletal muscle.

No information provided as to whether food intake was similar in mice consuming chow diet plus/minus simvastatin. Mice are notoriously sensitive to drugs added to diets.

How did the experimental interventions impact blood/plasma cholesterol lipid profiles?

Only GSH/GSSG ratio is reported. GSH and GSSG values should also be provided.

How are the complex I, II and IV activity data normalized? How do the authors know whether potential changes in liver mitochondrial content contributed to the findings? Is the decrease due to a change in enzyme expression, activity due to redox regulation, or loss of DHA? Lastly, electron transport complex activity is what is being measured, not "mitochondrial activity" as referred to in the text. Mitochondrial activity would imply mitochondria with intact mitochondrial membranes which, as shown in the Acin-Perez et al. reference, are not present in homogenates from frozen liver.

The relatively low n-size and seeming outliers in Fig. 5B does not instill confidence in the interpretation of these data. Related, wouldn't loss of one allele of Kdm5c in XX mice be expected to improve GT in the absence of statin treatment, at least according to the hypothesis put forward?

The experiments on iPSCs are difficult to interpret given that oxygen consumption in intact cells can be influenced by a multitude of factors.

Reviewer #4 (Remarks to the Author):

In this study Zhang and co-workers study the gender specific response to statins. The authors find that the mouse is a good model to study gender associated differences in response to statins. In subsequent studies they show that these effects involve a female specific higher susceptibility to glucose intolerance, fasting hyperglycemia, and muscle weakness after short-term statin treatment. Using the four-core mouse system they could pinpoint the X chromosome as driving this gender specific effect and show that Kdm5c as the crucial factor in this process. Finally, the authors extend and translate their findings to human. Overall, I find this a strong study with important implications for patients treated with statins. In addition, this study once more highlights the importance of studying effects in both genders. I have a few suggestions that may strengthen the manuscript.

-‘which are reduced by statin exclusively in female mice (see Fig. D).’ Which figure is referred to?

-‘When treated with statin, Kdm5c+/+ mice were susceptible to statin-induced glucose intolerance as we have seen throughout our studies with female mice, but Kdm5c+/- mice were protected (Fig. 5B).’

-Escape is never 100% so will be quite some heterogeneity with cells expressing a small reduction in Kdm5c expression (equal to that found in males) and others a much higher reduction (due to partial escape). This needs to be discussed.

-In that sense the following sentence is not right: ‘Thus, altering female Kdm5c gene dosage to that normally present in males largely prevents statin-induced impairment of glucose tolerance, redox tone, mitochondrial activity, and fatty acid gene expression.’ And needs explanation.

-Would be nice to have an indication of the Kdm5c expression level per cell.

-Is the action of Kdm5c direct or indirect. KDM5C ChIP-seq data is available and would strengthen the manuscript if the action is direct.

-With respect to the patient data, what is known about escape of KDM5C in the liver and other organs or cells (such as iPS cells) in women? This is important with respect to consistency of the postulated model. In addition, what could be the genetic determinant involved in NOD, it would be informative to study KDM5C expression levels in the iPSCs generated.

We thank the Reviewers for their thorough and constructive comments. We have addressed these to the best of our ability and modified the manuscript accordingly. Following is a point-by-point response to the critiques.

Reviewer #1 (Remarks to the Author):

In this manuscript, Zhang et al. report findings supporting previous observations suggesting women are more likely than men to experience statin-induced NOD and muscle related side effects. In a series of in vivo experiments and mechanistic studies, the authors show that short-term treatment with simvastatin resulted in worsening of glucose tolerance, increased fasting glucose and muscle weakness in female mice. Follow up studies identified that reduced levels of hepatic DHA, and impaired fatty acid metabolism, mitochondrial respiration, and redox balance, could be important contributing factors. This mechanism is further supported by analyses of gonadal sex and chromosomal sex differences that elucidate molecular and genetic determinants. When taken together, their findings suggest the potential for DHA supplementation to prevent statin-related side effects in woman.

The authors should be commended for investigating this important issue and for their efforts to uncover important sex differences in drug responses. While the findings linking statin-related side effects to altered lipid metabolism, mitochondrial function, and DHA are somewhat novel and interesting, there are several concerns related to the experimental approach and statistical analyses that affect the overall strength of the manuscript. The manuscript would also significantly benefit from additional mechanistic investigations that provide insight into the sex-specific actions of DHA in liver and how this is connected to skeletal muscle function.

Major concerns:

1. Statin dosing and exposure in in vivo models. In providing the rationale for in vivo model dosing, the authors reference two papers, Muraki et al., 2012 and Yokoyama et al., 2007. The studies of Muraki et al focus on atorva and prava, so it would appear the methodological rationale comes primarily from Yokoyama et al. Here the studies were conducted in a different genetic model in male mice only. Given the importance that a clinically relevant dose be studied, and perhaps more importantly, that there are not differences in drug exposure between sexes, the authors should demonstrate that dosing produces simva exposure equivalent to humans in their models and that no differences are observed between male and females. This is critical to interpret findings in the model and ensure that the observed effects do not arise from differences in PK. It would also be interesting to know whether statin liver exposures are different between sexes.

We appreciate the points raised by the Reviewer. There are hundreds of studies using simvastatin diets in the mouse, and **we selected simvastatin diet concentrations that are in line with a great deal of data in the literature** (typically simvastatin at 10–40 mg/kg/day, although there are many studies with much higher concentrations). Our concentrations amount to 0.1% simvastatin (drug weight/ food weight) mixed in chow, with a calculated intake per day of 16 mg simvastatin/kg body weight, which is equivalent to ~80 mg/day in humans (based on FDA conversion factor for mouse to human; details are provided in the Methods section of the manuscript). **We include additional representative citations** in the Methods section of the manuscript to support similar doses of simvastatin administered to mice. Please note, however, that extensive literature searches over the past several **years have not identified any studies that use dietary simvastatin and compare male and female mice. This underscores the contribution that our study provides in comparing the sexes and identifying differences in their response to statin.**

To the Reviewer's other comment, **we provide data in Figure 1A–C showing that statin added to the chow diet does not alter food intake or body weight gain in either females or males.** Additionally, our metabolomic data demonstrate that **statin led to reduced hepatic cholesterol or cholesterol ester levels in both sexes,** indicating that statin was having the expected effect (**Suppl. Fig. 2A**).

2. In figure 1, there appears to be differences in “chow” fasting glucose levels between male and female mice. Are they statistically different? This would be important to know when interpreting drug effects. Same for lipidomics, are there sex differences between “chow” treatment group?

We now display these male and female data on the same axes for easy comparison. The reviewer is correct that there are sex differences in basal glucose levels between males and females (now annotated in our figures), but only the females exhibit a statin-induced increase in glucose levels, and also have much early statin-induced glucose intolerance (after only 2 weeks) compared to males (at 8 weeks) (**Fig. 1C,D**). Regarding lipidomic analyses, there are differences between males and females in the basal levels of some of the most abundant fatty acids (C16:0, C18:0), but only females show decrements in fatty acid levels in response to statin (**Fig. 2C**).

3. More data are required to facilitate interpretation of results. Are there sex differences in body weight gain/body composition, food intake, etc.

Please see response to question 1 above. Our data show that the sex differences in statin response are not attributable to differential food intake or body weight gain.

4. Given that multiple tissues (including skeletal muscle, adipose, pancreas, etc) have been implicated in statin-induced NOS, it is not clear why the authors focused on liver lipid metabolism in the in vivo model. The impaired GTT could result from effects in one or more of these tissues. Evidence supporting a liver-specific mechanism is required as is a more thorough investigation into the other tissues.

We performed metabolomic profiling on liver, skeletal muscle, and plasma. As stated on page 4 and provided in Table S1, very minimal effects of statin treatment on metabolite concentrations were observed in either sex in muscle or plasma. However, statin did alter hepatic lipids and other metabolites with clear sex differences. We therefore focused on sex differences in hepatic lipid levels based on our data and because liver is the central organ for statin uptake, action, and metabolism.

Nevertheless, we agree with the Reviewer that sex-specific statin effects in skeletal muscle as well as liver may influence glucose homeostasis, especially given the impaired grip response of female mice treated with statin. Impaired glycogen storage could lead to alterations in glucose homeostasis. **In new experiments, we determined that females experience a 50% reduction in glycogen levels in both liver and muscle with statin treatment, whereas statin did not alter glycogen levels in male tissues (Fig. 3C, D).** Furthermore, **fish oil/statin co-therapy prevented the decrement in glycogen levels in liver and muscle of female mice,** and did not alter levels in males. The reduced glycogen levels in statin-treated females correlated with **reduced phosphorylation of glycogen synthase kinase 3 β (Fig. 3D),** which would correspond to reduced glycogen synthesis and storage. These data are consistent with sex-biased effects of statin on glucose storage as a potential contributor to sex differences in glucose homeostasis.

We acknowledge that it would be valuable to understand potential sex differences in statin effects in muscle, beta cells and other tissues, and we plan to do so in future studies. However, we believe that the current study identifies important sex differences in statin adverse

effects that are reflected in liver metabolism, and provides some of the contributing molecular and genetic determinants of these differences, as well as an effective therapeutic approach.

5. A deeper investigation into skeletal muscle is required to interpret the grip strength data. It is not clear how this fits into the working hypothesis.

We have assessed some additional aspects of statin response in muscle (note that all analyses mentioned here were performed with mouse quadriceps muscle to attain sufficient material). As mentioned earlier, our metabolomic analysis of muscle did not show significant differences due to sex or statin treatment (Suppl. Table 1). We now include data showing that **exclusively females experience a reduction in muscle mitochondrial DNA copy number in response to statin (Fig. 1G). We also provide new data showing a sex-biased effect of statin on glycogen levels in liver and muscle (Fig. 3C,D).** Please see response to question 4 for further description of these data.

6. The changes in hepatic fatty acid metabolism genes and metabolites (SF2 and F2C) appear to be robust and widespread.

a. Please provide a rationale for focusing specifically on DHA

The reviewer is correct that statin-treated females exhibit reductions in DHA and a few other fatty acids (C16:0, C18:1, C20:4). Of these, **DHA captured our attention because it sustained the greatest proportional reduction (40%), and because of known beneficial effects of omega-3 fatty acids on glucose homeostasis and muscle health.**

To provide further support for a role of DHA in preventing adverse statin effects, we have now assessed the effects of four fatty acid species that are reduced in statin-treated females on cultured hepatocytes treated with statin. As shown in **new Fig. 2E, cultured mouse hepatocytes exposed to statin (10 μ m simvastatin for 24 hr) had reduced viability that could be prevented by the addition of DHA, but not addition of saturated or monounsaturated fatty acids.** Although we do not intend to imply that the effects of DHA in vivo are necessarily on cell survival, these data are in agreement with our in vivo data. **Ultimately, the demonstration that providing omega-3 fatty acids via fish oil supplementation during statin treatment prevents adverse effects in females serves as a proof of principle that this is a key component. Please note that in these studies, providing non-omega-3 fatty acids (as was done in the control animals, who received coconut oil) did NOT prevent statin adverse effects.** (Please also see our lipidomic analyses of fish oil and coconut oil in Suppl Fig. 3).

b. Are there differences in liver, muscle, and plasma TG mass? Sex differences?

Our lipidomics data show no differences in hepatic triglyceride levels between males and females, nor between animals on chow or chow plus statin (the primary TG species are shown below).

7. Please provide rationale for using atorva vs simva for iPSC experiments and why tested at 250uM, a conc. orders of magnitude above free drug levels in humans

Atorvastatin and simvastatin were the most prevalent statin types prescribed to the donors of the iPSCs. To determine optimal statin dose to use for short-term treatment (24hr) in cultured iPSCs, we performed extensive dose response curves with simvastatin and atorvastatin in five randomly selected human iPSC lines. From these analyses (shown below, each colored line represents a different iPSC line) we identified 500nM simvastatin, and 250nM atorvastatin as the lowest statin concentrations that would yield robust and reproducible induction of *HMGR* and reduction of *MYLIP*, two well established transcriptional changes that occur in response to statin treatment. While lower concentrations of these statins did produce a reliable up-regulation of *HMGR*, consistent with activation of SREBP2 in response to reduced intracellular cholesterol levels, the lower doses did not yield consistent reductions in *MYLIP*. Thus, for studies in the manuscript, we settled on treating the iPSCs with atorvastatin because it was effective at a lower dose. We suspect that this is because atorvastatin is administered in its active form, whereas simvastatin is introduced as an inactive lactone derivative.

While the statin concentrations used in vitro are higher than in vivo, it is important to note that iPSCs do not express many of the known statin transporters, such that higher concentrations are required to bring about the normal response in gene expression. Additionally, it is difficult to extrapolate from an acute dose (24 hr) compared to the chronic statin exposure of patients (years). Notwithstanding these complications of reducing an in vivo state to a cell culture system, the differences in sex and disease context that we reported were obtained from a constant set of conditions across all samples.

Minor concerns:

1. Please report AUC data for figures 1a&b and 2e&F- how they are calculated and statistical methods

These data are now included in Figure 1, and we have moved the GTT curves to Suppl. Fig. 1. The AUC under the GTT curves was determined using trapezoidal analysis of blood glucose values as described (Solberg LC et al. (2006) *Mamm Genome*), which is now cited in the Methods section.

2. Please explain discordance between grip strength results for figures 1 and 2. Did statins impair grip strength in F2e? If not, what is the interpretation of the fish oil results.

We apologize for the confusion. The reviewer presumably noted that the absolute grip strength values differ between experiments shown in Fig. 1F and those in 3B and 5C. This is due to the fact that the initial studies in Fig. 1 were performed with a different grip meter than we used for subsequent experiments. During the course of our studies, we refined our measurement capabilities by developing a grip meter with a microcontroller (Munier et al. (2022) *Sci Rep*), which we used for subsequent experiments. We now make this transparent in the Methods section.

Reviewer #2 (Remarks to the Author):

The manuscript entitled “X chromosome dosage drives statin-induced dysglycemia and mitochondrial dysfunction” by Zhang and colleagues, brings some experimental evidences in mice and humans, to explain why woman are more susceptible to the side effects on muscle and glucose homeostasis induced by statin treatment.

Please find below my main comments:

- Literature about gender and myopathy is not really solid and the Authors only cited one paper that support this idea. This cause-effect relationship doesn't seem to be a stablished concept.

Overall, there is a dearth of data that directly compare statin adverse effects in men and women. Since we first submitted this manuscript, additional data showing sex bias in statin-related myopathy have provided further evidence for a female bias in statin adverse effects. In the most comprehensive study to date, a meta-analysis of 176 statin trials with more than 4 million subjects identified female sex as the strongest risk factor for statin intolerance, including myopathy/muscle pain as a major outcome (female sex, statin intolerance OR 1.47, $p = 0.007$) (Bytyci et al. (2022) *Eur Heart J*). Female sex had a greater impact than age, ethnic background, obesity, diabetes, hypothyroidism, or chronic liver or renal failure. We acknowledge that there may be variation in the effects reported in individual studies (which could be due to differences in genetic background, control for environmental variables, and methodological differences between studies), but the data from the very large meta-analysis are quite compelling that sex plays a key role. We now cite this in the Introduction.

- The Authors should have add a group of hypercholesterolemic mice treated with statins to try to reproduce what happens in humans who are treated with statins.

We apologize if it was not clear, but we did include a mouse cohort with hyperlipidemia: the Four Core Genotypes mice in Fig. 5 are apoE-deficient, which renders them hypercholesterolemic on a standard chow diet. The lipid levels for this particular cohort were presented in a separate publication (Wiese et al. 2022), and so they are summarized in the text together with the relevant citation. The female-biased occurrence of statin-induced glucose intolerance and impaired grip strength are observed in these animals, consistent with the findings in non-hypercholesterolemic mice elsewhere in our study. Thus, these statin-intolerant phenotypes occur under either normo- or hypercholesterolemic conditions.

- The poor metabolic phenotyping in Figure 1 is an evident shortcoming of this study. The Authors do not provide data on insulin resistance, although some degree of high glucose concomitant with blood insulin indicates the females might be insulin resistant. Does that match with the literature?

We apologize if it was not clear, but we provided insulin levels in males and females treated with statins for 4, 8, and 16 weeks in Suppl. Fig. 1. We now also include HOMA-IR calculations in the same supplementary figure, and comment in the text. The findings indicate that statin treatment did not alter insulin levels or HOMA-IR in male or female mice, but male mice had higher absolute insulin levels and HOMA-IR compared to females. We could not find any studies in the literature that show effects of statin treatment on female mice on glucose parameters, so cannot comment on existing literature.

- GTT experiments reflects changes in glucose uptake too, but mainly reflects the glucose induced insulin secretion. There is no dosage of insulin along the glucose dosages that may explain the results.

Unfortunately, we did not have sufficient plasma from glucose tolerance tests to assess insulin at time points. This would require an additional dedicated mouse cohort, which is not currently available. Future studies will delve into this further by performing dedicated studies using clamps.

- Elevated fasting glucose may reflect an enhanced hepatic glucose production which was not evaluated here. Many other experiments should have been provided, such as gene/protein expression of gluconeogenic genes and proteins, pyruvate test, etc. Moreover, the GTT experiments at Fig.1 do not display the AUC or statistical analysis, albeit it seems that the intolerance was observed in both genders. Lack of statistical analysis and AUC for the GTTs makes the interpretation of the data a little difficult.

We apologize if it was not clear, but we did indicate statistical significance for AUC of GTT curves in Fig. 1 via asterisks for samples that showed a significant area under the curve (denoted as AUC* or AUC** for $p < 0.05$ and $p < 0.01$, respectively). However, to make things clearer, we have now moved the AUC denoted as bar graphs from Suppl. Fig. 1 to main Fig. 1, and moved the glucose curves from the main figure to Suppl. Fig. 1. We hope that the reviewer can now better appreciate that the females experience glucose intolerance in response to statin after only 2 weeks of treatment, whereas males do not develop glucose intolerance until 8 weeks. Furthermore, only the females developed elevated fasting glucose levels, even after 16 weeks of statin treatment.

We have also added new data in Fig. 3 that shows that both liver and muscle from females, but not males, experience about a 50% reduction in glycogen levels in response to statin; this is ameliorated when fish oil is provided with the statin. Furthermore, the reduced glycogen levels are associated with reduced phosphorylation of GSK-3 β , which is consistent with reduced glycogen synthesis.

We have analyzed expression levels for gluconeogenic genes and do not see sex differences nor effects of statin, and did not include these in the manuscript, but provide below for the reviewer.

We acknowledge that it would be valuable to characterize the glucose homeostasis further by techniques such as clamp studies and we hope to do so in the future. However, we maintain that the current study contains a great deal of data and insight into sex differences in statin adverse effects. These include the identification of contributing genetic (X chromosomal dosage, and specifically dosage of the X escape gene, *Kdm5c*) and molecular determinants of these differences (reduced omega-3 FA levels in both female mice and humans, which are associated with impaired Nrf2 signaling and redox tone, impaired mitochondrial activity, etc.). Our study also demonstrates an effective therapeutic approach to prevent statin adverse effects in female mice.

- It is not clear in text how the Fatty acids quantified in the Figure 2C are in the free form or it is a compilation of the FAs incorporated in the different forms (i.e. as LPC, CL, PC, TAGs, DAGs, etc...). A more detailed presentation of the lipidomic data should be provided.

Fatty acids were quantified in the free form. The complete lipidomic data is presented in Supplemental Tables.

- It seems that all the long-chain FAs showed in Fig 1C, including saturated and monosaturated ones, are reduced. The lack of this pool of FAs can lead to the malfunction of mitochondrial machinery, since this result in a lack of FA substrates for fatty acid oxidation pathway in mitochondria and consequent lack of further generation of intermediates for the OXPHOS at electron transport chain.

This is a good point, and we agree that reduced fatty acid substrates could contribute to the impaired mito respiration that we determined in liver of statin treated female mice. However, we observe impaired mitochondrial respiration in female mice treated with statin in conjunction with coconut oil which is rich in fatty acid substrates for oxidation (these are the mice treated with stat and vehicle in Fig. 4 G and H). Furthermore, the studies of mitochondrial activity in cultured human iPSCs would likely not be subject to differences in fatty acid levels as they were all cultured for multiple passages in culture medium replete with fatty acids.

- The Authors did not provide the composition of the fish or coconut oil, although it only indicates the total amount of DHA and EPA included. However, several other FAs are being provided together with coconut and fish oil, and it is important to know which ones and how much of

them. Moreover, it doesn't seem that the circulatory levels of DHA and EPA was provided after fish oil treatment. Did the treatment restore the DHA levels in the liver?

We have now **performed lipidomic analysis of the fish oil and coconut oil that we used in our studies in Suppl. Fig. 3**. These analyses confirmed that the fatty acid composition of the two oils differed primarily in the levels of omega-3 fatty acids (DHA, EPA, and docosapentaenoic acid, C22:5), although there were differences in a few other species as well (C12:0, C18:3 and C20:3), but these were actually higher in the coconut oil. **The two oils were very similar in their levels of the abundant medium and long chain fatty acid species aside from omega-3 fatty acids**. Also, please note that in our experimental design, the fish oil was given in combination with coconut oil (1:4 vol:vol), so all animals received a nearly similar amount of coconut oil.

- The link between the mitochondrial alterations and the dysglycemia is not clear, since there no mechanistic demonstration that clarify this hypothesis.

This is obviously a complex question, but our data are consistent with previous work that link redox status to glucose homeostasis. Our data indicate that statin adverse effects are associated with a reduction in glutathione levels. It has previously been shown that abnormal glutathione redox status leads to reduced mitochondrial complex I activity [e.g., Jha et al. (2000) *J Biol Chem*] and to reduced insulin sensitivity [e.g., De Mattia et al. (1996) *Metabolism* 47:993–996; Paolisso et al. (1992) *Am J Physiol* 263:E435440; Azarova et al. (2023) *Int J Mol Sci* 24:4738]. We add this to the Discussion section on page 12.

- CoQ10 is a key component of the mitochondrial electron transport system. Therefore, CoQ10 deficiency resulting from statin treatment may impair cellular energy metabolism, and contribute to the development of myopathy and muscle symptoms, as described in patients treated with statins in many previous studies (<https://doi.org/10.1016/j.jacc.2007.02.049>, doi: 10.1023/B:CARD.0000015861.261111.ab., doi: 10.1001/jama.289.13.1681., doi: 10.1002/biof.5520180221). It is very likely that CoQ10 deficiency is the key link between statins treatment and the results observed in the females, CoQ10 should be also measured in the other conditions including the knockout mice and fish-oil treated mice. There are evidences that DHA may increase CoQ10 (doi: 10.1111/mcn.12300 doi.org/10.1016/j.redox.2021.102061).

We are aware of the link between CoQ10 levels and statin adverse effects, and acknowledge that this may contribute to statin adverse effects in some individuals and experimental models. However, in our system in C57BL/6 mice, **males and females showed similar CoQ10 levels, and did not exhibit significant differences in CoQ10 levels in response to statin** (please see Suppl. Fig. 2A). **Given that only females exhibit the statin intolerant phenotypes that we examined, it is not possible to attribute them to reduced CoQ10 levels in females.**

We fully acknowledge that the response to statins and their detrimental effects are clearly complex and multifactorial, and will likely depend on sex, genetic background, age, and likely many other factors. Thus, we do not intend to imply that the mechanisms underlying sex differences in statin adverse effects that we identified represent the only perturbations that may influence statin intolerance as these may be influenced by not only sex but also genetic background, age, type of statin, etc. Nevertheless, our findings provide definitive evidence for a sex difference in statin intolerance in mice that are otherwise genetically and experimentally identical, identify a key biochemical component (omega-3 fatty acid) and genetic component (an X chromosome gene) that can each prevent the statin adverse effects in females, and also demonstrate sex differences in omega-3 fatty acid levels and mitochondrial function in human statin recipients.

Reviewer #3 (Remarks to the Author):

The study by Zhang et al. explores the potential mechanisms by which women experience adverse effects to statin therapy relative to men. The experiments provide a number of interesting findings, including that female mice on simvastatin therapy are characterized by glucose intolerance, fasting hyperglycemia, lower grip strength, decreased DHA levels, decreased hepatic GSH/GSSG ratio, and decreased electron transport complex I, II and IV enzyme activities. Interestingly, supplementing statin-treated female mice with fish-oil prevents the adverse effects. The data is presented logically and the manuscript well-written.

The main concern with the study is that the effect of statin and statin counter therapies were evaluated in lean chow fed mice, not in the context of obesity, western diets, and/or hypercholesterolemia. It is not clear why potential differences in statin adverse effects between sexes were not studied under more translationally relevant conditions. While the argument can be made that chow fed mice represent a more controlled condition, it is entirely possible that the sex differences identified are specific to the context in which they were studied and therefore of limited translational relevance.

Other areas for potential improvement include the following:

Assessment of muscle function/structure/myopathy limited to grip strength. The study could be strengthened considerably by analysis of skeletal muscle.

We agree that this is important, but also that our study provides numerous aspects of the genetic and molecular basis for sex differences in statin adverse effects, as well as an effective therapeutic strategy, even if not all aspects of the response in each tissue are understood at present. Our metabolomics data in muscle did not show differences between the sexes that could provide the basis for hypothesis testing in the way that the sex differences in liver did. We have sought clues for differences in muscle and now include data that show female muscle mitochondrial content is lower than that in males, and this is diminished further in response to statin (Fig. 1G). Additionally, we found that statin leads to reduced muscle glycogen storage as well as reduced hepatic glycogen storage specifically in females (Fig. 3C). Further understanding of the basis for sex differences in statin response in muscle will require dedicated focus on this issue, but does not detract from the large amount of genetic and molecular insight that our studies provide.

No information provided as to whether food intake was similar in mice consuming chow diet plus/minus simvastatin. Mice are notoriously sensitive to drugs added to diets.

We now provide food intake and weight gain data for both sexes on chow and statin-containing diets (Fig. 1A–C).

How did the experimental interventions impact blood/plasma cholesterol lipid profiles?

The goal of the studies here was to investigate statin adverse effects, and most of the experiments were performed in mice on a chow diet, which therefore already had low lipid levels. Please also see response to Reviewer 2.

Only GSH/GSSG ratio is reported. GSH and GSSG values should also be provided.

We now report the values for all glutathione forms in Suppl. Fig. 4, including GSH, GSSH, and GSH/GSSG ratio.

How are the complex I, II and IV activity data normalized? How do the authors know whether potential changes in liver mitochondrial content contributed to the findings? Is the decrease due to a change in enzyme expression, activity due to redox regulation, or loss of DHA? Lastly, electron transport complex activity is what is being measured, not “mitochondrial activity” as referred to in the text. Mitochondrial activity would imply mitochondria with intact mitochondrial membranes which, as shown in the Acin-Perez et al. reference, are not present in homogenates from frozen liver.

Mitochondrial complex activities were determined in liver protein homogenates and normalized to liver protein levels. We have no reason to suspect that statins affect overall protein levels, so this normalization is very reasonable. We apologize for referring to our measurements as “mitochondrial activity” and have revised to indicate that we are actually determining electron transport complex activity.

The relatively low n-size and seeming outliers in Fig. 5B does not instill confidence in the interpretation of these data. Related, wouldn't loss of one allele of *Kdm5c* in XX mice be expected to improve GT in the absence of statin treatment, at least according to the hypothesis put forward?

Obviously, greater numbers of animals are always desirable, but it is difficult to attain *Kdm5c*^{+/-} mice on an inbred C57BL/6 background; many appear to perish in utero, although those that survive appear to be relatively healthy. Please note that in our 2-way ANOVA that is performed on all studies in Fig. 6 prior to gaining permission to perform pairwise comparisons, we gain power because the groups are paired up for the analysis (i.e., all +/+ vs. all +/- animals, as well as all chow vs. all statin animals). For comparisons that gave a significant 2-way ANOVA result or indicated a significant interaction between genotype and treatment, pairwise comparisons were performed and many found to be significant. The consistency of the results for multiple phenotypes also provides confidence.

We would NOT expect that *Kdm5c*^{+/-} mice fed a chow diet will have improved glucose tolerance—none of the mice a glucose intolerant under chow conditions. It is possible that if we made these mice hyperlipidemic we might see an effect of the *Kdm5c* gene dosage without statins, but here we were focused on assessing adverse statin effects.

The experiments on iPSCs are difficult to interpret given that oxygen consumption in intact cells can be influenced by a multitude of factors.

Respectfully, we are unclear what the Reviewer is specifically critiquing in these experiments. Certainly many factors influence mitochondrial oxygen consumption. In our studies, **all of the patient-derived iPSCs are cultured, treated, and analyzed under identical conditions**, such that the main variable is the characteristics of the individual from whom the cells are derived, with the relevant factors here being sex and response to statin (susceptibility or resistance to NOD). **The data are clear cut in demonstrating that with identical culture conditions, statin treatment of iPSCs from women with NOD leads to reduced oxygen consumption, whereas iPSCs from women who did not develop NOD when taking statins or from men did not experience impaired oxygen consumption.**

Reviewer #4 (Remarks to the Author):

In this study Zhang and co-workers study the gender specific response to statins. The authors find that the mouse is a good model to study gender associated differences in response to statins. In subsequent studies they show that these effects involve a female specific higher susceptibility to glucose intolerance, fasting hyperglycemia, and muscle weakness after short-term statin treatment. Using the four-core mouse system they could pinpoint the X chromosome as driving this gender specific effect and show that *Kdm5c* as the crucial factor in this process. Finally, the authors extend and translate their findings to human. Overall, I find this a strong study with important implications for patients treated with statins. In addition, this study once more highlights the importance of studying effects in both genders. I have a few suggestions that may strengthen the manuscript.

-‘which are reduced by statin exclusively in female mice (see Fig. D).’ Which figure is referred to?

Apologies for the error; this has been corrected.

-‘When treated with statin, *Kdm5c*^{+/+} mice were susceptible to statin-induced glucose intolerance as we have seen throughout our studies with female mice, but *Kdm5c*^{+/-} mice were protected (Fig. 5B).

We are uncertain what specific critique the reviewer is making here. The point is that wild-type female mice (*Kdm5c*^{+/+}) develop glucose intolerance when treated with statin, but females in which one allele has been inactivated to give them the same gene copy number as males (*Kdm5c*^{+/-}) are protected (please see Fig. 6B).

-Escape is never 100% so will be quite some heterogeneity with cells expressing a small reduction in *Kdm5c* expression (equal to that found in males) and others a much higher reduction (due to partial escape). This needs to be discussed.

-In that sense the following sentence is not right: ‘Thus, altering female *Kdm5c* gene dosage to that normally present in males largely prevents statin-induced impairment of glucose tolerance, redox tone, mitochondrial activity, and fatty acid gene expression.’ And needs explanation.

By ‘gene dosage’ we are referring to the number of functional *Kdm5c* gene copies. By ablating one *Kdm5c* allele, we are converting XX females with two functional *Kdm5c* alleles to XX females with 1 *Kdm5c* allele, which is the normal gene dosage in XY individuals. However the Reviewer is correct that there will be two levels of *Kdm5c* expression within tissues of *Kdm5c*^{+/-} mice and we have added text to the Discussion that discusses this (page 12). It would require single cell RNA-seq to identify the consequence of *Kdm5c* haploinsufficiency at the cellular level, which is outside of the scope of this study.

We wish to reemphasize that this study has taken us all the way from the demonstration that male and female mice (genetically identical except for sex) exhibit differences in statin tolerance, to the identification of associated sex differences in lipidomic, transcriptomic, and metabolic responses to statin, to a demonstration that X chromosome dosage is a determinant, and finally, to identification of the involvement of a specific X chromosome gene (*Kdm5c*). We further demonstrate a therapeutic lipid (omega-3 fatty acids) that prevents statin intolerance in mice, and which shows reduction in women (but not men) taking statin as well. We hope that

the reviewer appreciates that this study therefore adds a lot to what was previously known about sex differences in statin tolerance, and provides a gene candidate for future study.

-Is the action of Kdm5c direct or indirect. KDM5C ChIP-seq data is available and would strengthen the manuscript if the action is direct.

We agree with the Reviewer that it will be exciting to identify KDM5C target genes that underlie our finding that *Kdm5c* gene dosage influences statin intolerance (a point we bring up in the Discussion). At present there are no ChIP-seq data for KDM5C in liver or hepatocytes (or other relevant cell types) available in the public epigenomic datasets (ENCODE, ChIP-Atlas). Therefore these are datasets that we will need to generate ourselves in future studies, and will be much more meaningful as we can compare effects of statin treatment as well.

-With respect to the patient data, what is known about escape of KDM5C in the liver and other organs or cells (such as iPSCs) in women? This is important with respect to consistence of the postulated model. In addition, what could be the genetic deterrent involved in NOD, it would be informative to study KDM5C expression levels in the iPSCs generated.

As we have presented in earlier work and cite in the manuscript, the KDM5C gene escapes X inactivation across mammalian species, including humans. This is true across key metabolic tissues including liver, adipose tissues, skeletal muscle, etc. This is nicely demonstrated in data from human tissues in GTEx. Shown below are the *KDM5C* expression levels across 40 human tissues separated by sex (women in pink, men in blue), showing higher expression levels in women for most tissues, including liver. For comparison, we show an X-linked mitochondrial respiratory chain gene *COX7B* (which does not escape inactivation), which shows comparable expression levels between men and women.

To address the Reviewer's comment about *KDM5C* levels in the iPSCs, we now include a new Fig. 8D that shows the expected elevated expression of *KDM5C* in iPSCs from women compared to men, as well as expression of an X chromosome gene that does not escape X inactivation, which therefore has similar levels in men and women.

REVIEWER COMMENTS

Reviewer #1 (Remarks to the Author):

The manuscript is greatly improved. I do not have additional comments.

Reviewer #2 (Remarks to the Author):

Most of the points raised by this Reviewer were responded accordingly. A few points are still a concern.

- Changes in glucose homeostasis are not really consistent.
- Differences in fasting glucose are just discrete and do not reproduce in the GTT experiments as showed in the supplementary figures. It seems that gluconeogenesis pathway is not altered. More investigation on glucogenolysis pathway could be done.
- And the differences in GTT are not really convincing.
- Authors must revise the qPCR calculations. In fig 6E it seems that Acaca and Fasn gene expression is relative to the WT statin group and not by WT chow group as it should be. Elovl6 is correct.
- In Figure 8D, Kdmc5 expression is not importantly regulated in iPSC from females in comparison to males. The Authors zoomed the Y axis from 11.5 to 13.5. Why did the authors did not expressed this data as relative mRNA expression as previously done in Fig 6E? Authors should provide protein expression data.

Reviewer #3 (Remarks to the Author):

The authors have responded adequately to my concerns.

Reviewer #4 (Remarks to the Author):

The authors have addressed all my questions/concerns.

Response to review. In the previous revision, 3 of the 4 referees indicated that all critiques had been addressed and they had no concerns. A single reviewer added additional critiques, which we address here.

Reviewer 2 Comments:

Most of the points raised by this Reviewer were responded accordingly. A few points are still a concern.

- Changes in glucose homeostasis are not really consistent.

RESPONSE: It is unclear what the reviewer is referring to. We hope the reviewer understands that *we are focused on differences between the sexes in the change in fasting glucose and glucose tolerance in response to adding statin to the chow diet, rather than a comparison between absolute values for males and females.* (It is well documented that males have higher fasting glucose levels than females, and our data on the chow diet are in agreement with this. Also note that all mice are fed a chow diet—there is no high-fat feeding in the study, which is required to cause larger elevations in glucose levels that the reviewer may be accustomed to seeing.) *Consistently across 3 different mouse models, we show that genetically female mice are more susceptible to statin-induced glucose intolerance than genetically male mice:*

(1) In cohorts of female and male C57BL/6J mice across progressively longer time points for statin treatment (2 weeks, 4 weeks, 8 weeks, and 16 weeks), females developed impaired glucose tolerance earlier and to a greater extent than males (**Fig. 1**).

(2) In the Four Core Genotypes mouse model, statin-induced glucose intolerance occurred in XX but not XY mice (**Fig. 5**).

(3) In a model with altered copy number of the X chromosome gene, *Kdm5c*, having the female dosage of this gene (2 copies) leads to glucose intolerance compared to having the male dosage (1 copy) on an otherwise female background (**Fig. 6**).

- Differences in fasting glucose are just discrete and do not reproduce in the GTT experiments as showed in the supplementary figures. It seems that gluconeogenesis pathway is not altered. More investigation on gluconeogenesis pathway could be done.

RESPONSE: The fasting glucose levels in our study are all taken after a fast in a specific time frame (0800–1300) which is analogous to how fasting glucose levels are determined in humans; these always result in a discrete value. The reviewer is incorrect in stating that these values do not agree with the GTT experiments, as the fasting glucose levels come from the 0 time point of the GTT assays.

In the first revision of this manuscript, we added data to demonstrate that *the defect that underlies elevated glucose levels in muscle and liver of females in response to statin is impaired glycogen synthesis (Fig. 3C,D, shown below for the reviewer's reference).* Specifically, we showed that the glycogen storage levels and activation status of the glycogen synthesis regulator, GSK-3 β , are impaired in liver and muscle from statin-treated female (left), but not male (right), mice. We also showed that the defect in glycogen storage and GSK-3 β regulation is corrected by statin co-therapy with fish oil.

In response to the reviewer's new comment, we have added Suppl. Fig. 4 (copied below). We find no evidence of increased gluconeogenesis nor increased glycogenolysis in statin-treated females. In fact, expression of the first enzyme in the hepatic glycogenolysis pathway (glycogen phosphorylase, *Pygl*) is *down-regulated* in statin-treated females, which would result in *reduced* glycogenolysis.

Supplemental Figure 4

- And the differences in GTT are not really convincing.

RESPONSE: The effects of statin on reducing glucose tolerance in females is *statistically significant in every mouse cohort tested* (see also response to the first comment, above). Perhaps the reviewer does not realize that *all mice are fed a chow diet (not a high-fat diet) and therefore are not expected to have glucose intolerance at all. It is quite notable, therefore, that simply adding statin to the chow diet results in impaired glucose tolerance.*

- Authors must revise the qPCR calculations. In fig 6E it seems that Acaca and Fasn gene expression is relative to the WT statin group and not by WT chow group as it should be. Elovl6 is correct.

RESPONSE: We agree with the reviewer and apologize for this oversight. We have corrected the graphs.

- In Figure 8D, Kdmc5 expression is not importantly regulated in iPSC from females in comparison to males. The Authors zoomed the Y axis from 11.5 to 13.5. Why did the authors did not expressed this data as relative mRNA expression as previously done in Fig 6E? Authors should provide protein expression data.

RESPONSE: We appreciate the reviewer's comment and have replotted the gene expression data from human iPSCs. The original plots were shown as log2 fold-change, which had an effect of compressing the gene expression values on the y-axis. We have now plotted the data as actual expression levels (not log transformed), which makes the differences in *KDM5C* expression levels between men and women more visually apparent. Note that *the difference in KDM5C expression levels between men and women is highly statistically significant, whereas X chromosome genes that do not escape X inactivation (such as FLNA) do not differ between the sexes* (see Fig. 8D below).

REVIEWERS' COMMENTS

Reviewer #2 (Remarks to the Author):

This Reviewer is in agreement with the response sent by the authors.